# Financial markets value skillful forecasts of seasonal climate

Derek Lemoine [1,2] ✉ & Sarah Kapnick[3,4]

Scientific agencies spend substantial sums producing and improving forecasts of seasonal climate, but they do so without much information about these forecasts' value in practice. Here we show that financial market participants value the production of seasonal forecasts: options traders price the uncertainty generated by upcoming United States National Oceanic and Atmospheric Administration Winter and El Niño Outlooks. Each outlook affects firms throughout the economy, with total market capitalization of $6 and $13 trillion, respectively. A 1% improvement in the skill of the El Niño Outlook reduces firms' exposure to a one standard deviation shock by $18 billion and induces traders to spend an additional $2 million hedging the outlook's news. Firms must not be able to undertake ex-ante adaptation that would eliminate their exposure to the forecasted portion of seasonal climate without imposing substantial costs of its own.

Governments spend substantial resources funding scientific agencies to produce forecasts of seasonal climate (defined as weather two weeks to one year ahead)[1]. The United States (U.S.) Weather and Research Forecasting Innovations Act of 2017 elevated seasonal forecasting innovations to one of the National Weather Service's five focus areas[2], and the European Centre for Medium-Range Weather Forecasts' 2021–2030 strategy highlights producing skillful seasonal outlooks as one of the four outcomes that indicate progress in meeting user needs[3]. Yet water resource managers have not relied on seasonal forecasts and would not prioritize their further improvement[4,5], and there are theoretical reasons to believe that the existence of skillful short-run forecasts undercuts the value of longer-run seasonal forecasts[6]. Many have lamented that policy priorities must be developed without knowing society's current value for forecasts or how that value would increase if forecasts became more skillful[7–9].

The ideal approach to valuing forecasts would observe people directly revealing their values with real bets in markets[10–13]. Previous work instead valued forecasts by computing the value of information within a model of some particular decision problem[14–16], by attributing all unexplained volatility in financial markets to weather risk[17], or by surveying assumed users directly[18]. Other work showed that financial

or prediction market participants attend to more conventional weather forecasts that have horizons of days[19–23] and to climate model forecasts of multi-year trends[23]. It is an open question whether traders also attend to seasonal forecasts that are less well-known, less skillful, less precise, and less immediately relevant than short-run weather forecasts. Learning whether traders do indeed value longer-run forecasts and which economic sectors they see seasonal forecasts affecting should inform how governments allocate funds towards producing and improving seasonal forecasts.

We observe that traders of options on firms' stocks must regularly place implicit bets on the contents of multi-month seasonal outlooks if these outlooks might affect firms' expected profits. Financial options provide their holders with the right—but not the obligation—to either buy or sell an underlying asset at a predefined "strike" price by some expiration date. An option's holder gets to profit from favorable price movements but gets to walk away from unfavorable price movements. Because options provide unlimited upside risk but limited downside risk, they become more valuable as uncertainty about the underlying asset's future price increases[24,25]. Options markets should reflect the uncertainty induced by an upcoming seasonal outlook if its possible forecasts would affect firm value and, thus, stock prices. They should

[1]University of Arizona, 1130 E. Helen St, McClelland 401, Tucson, AZ 85721, USA. [2]National Bureau of Economic Research, 1050 Massachusetts Ave, Unit 32, Cambridge, MA 02138, USA. [3]National Oceanic and Atmospheric Administration, 1401 Constitution Avenue NW, Room 5128, Washington, DC 20230, USA. [4]Formerly Geophysical Fluid Dynamics Laboratory, National Oceanic and Atmospheric Administration, Princeton, NJ 08540, USA. ✉e-mail: dlemoine@email.arizona.edu

not reflect the uncertainty induced by an upcoming seasonal outlook if traders do not judge the outlook to be skillful, if traders judge seasonal climate to be irrelevant to profits, or if the outlooks' long lead times allow firms to cheaply minimize exposure to the forecasted seasonal climate.

We test whether options markets in 2010–2019 priced uncertainty about the news contained in upcoming seasonal climate outlooks for winter weather, the El Niño Southern Oscillation (ENSO), and hurricanes. The U.S. National Oceanic and Atmospheric Administration (NOAA) releases each seasonal outlook on a regular, announced schedule with strict rules to guard against information leaking in advance (see Supplementary Discussion 2). Each product provides multi-month predictions (and hence is a "seasonal" outlook). "Skill" refers to the accuracy of an outlook's forecasts over many years. Supplementary Table 2 collects the outlooks' release dates. The Atlantic Hurricane Seasonal Outlook is released in May and includes outlooks for both the Atlantic and Eastern Pacific basins. The U.S. Winter Outlook is typically released on the third Thursday of October. It reports the probability that each part of the country will experience abnormal seasonal temperatures or precipitation over the coming December through February. The ENSO seasonal outlook is released monthly. ENSO refers to sea surface temperature and wind anomalies in the eastern Pacific. The state of ENSO is often found to predict climate variables elsewhere in the world, including temperature and precipitation[26]. Each month's outlook reports the current state of ENSO and provides predictions out to 9 months. The June ENSO Outlook is known to be especially informative because it is the first to take advantage of the jump in skill after the "spring barrier" (See Supplementary Table 1)[27–32]. This jump in skill arises in part because contributing factors to ENSO are especially noisy in the spring[33].

Figure 1 illustrates our methodology, which we detail in the Methods. The price of a stock is its ("risk-neutral") expected value across possible seasonal climate outcomes. In this example, that price may jump to $50 or $70 upon the release of a seasonal outlook but

should not change on average. Testing for these stock price reactions would reveal whether a particular year's forecast was both surprising relative to expectations and relevant for profits. However, stock price reactions do not tell us how markets value the regular production of outlooks because different years' outlooks should not affect stock prices on average. Some prior work studies effects of forecasts in futures markets;[19–21,23] for our purposes, futures prices act like stock prices, as they average over possible future outcomes. Further, with either stocks or futures, it is hard to increase power by combining information from multiple securities because opposing effects across firms (as in ref. 34) or commodities could cancel each other out.

In this work, we show that traders' uncertainty about future stock prices tends to fall upon the release of an outlook. In Fig. 1, the standard deviation of future stock prices is smaller once the outlook is released. Even if some particular outlook's release happened to increase traders' uncertainty by forecasting an especially volatile climate, the law of total variance implies that releasing outlooks should reduce their uncertainty on average (see Supplementary Discussion 1). We measure traders' uncertainty from the standard deviation implied by option prices, commonly referred to as "implied volatility" (see Methods). If markets anticipate outlook releases, then each affected firm's implied volatility should, on average, decline when outlooks are released. We increase our power to detect an effect in our base analysis by combining information from thousands of firms' responses and ten years of outlook releases within an event study regression framework. The regression controls remove the effects of news related to factors such as interest rates and earnings reports. We test whether the residual movement in implied volatility on the days a seasonal outlook is released is unusual relative to other days in the sample (see Methods).

## Results
### Outlooks affect markets throughout the economy
Figure 2a plots the average effect of NOAA outlooks on implied volatility across all firms that have liquidly traded options on U.S. equity

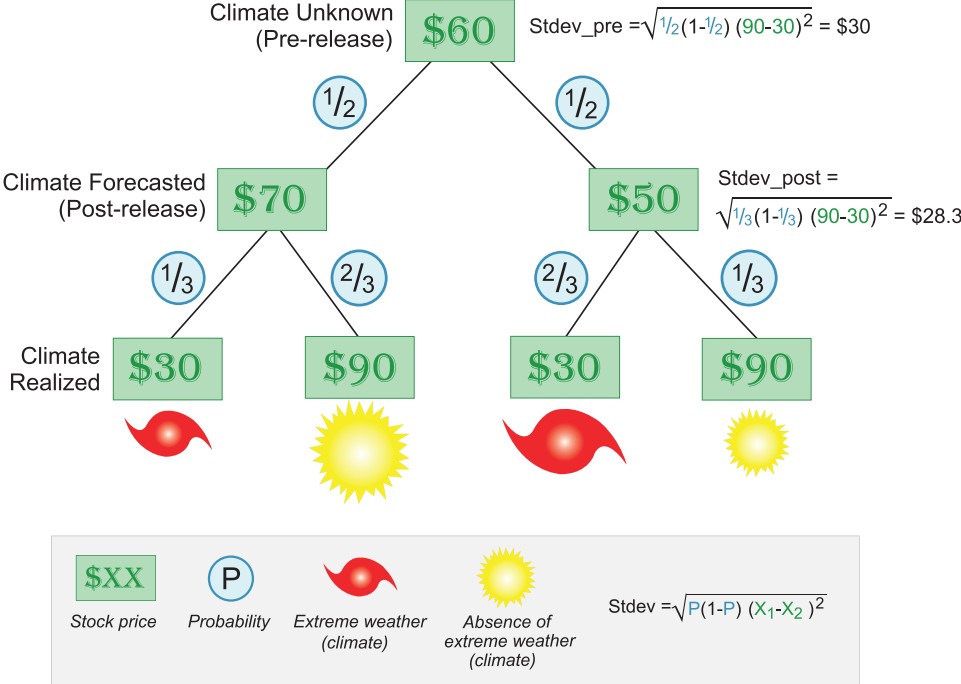

**Fig. 1 | Illustration of why options capture the value of seasonal outlooks.** This example depicts a seasonal climate that can have only two possible weather outcomes (extreme weather or absence of extreme weather, leading to a stock price of either $30 or $90) and an outlook that will provide one of two, equally likely forecasts (leading to a stock price of either $70 or $50). Stdev_pre is the standard deviation of stock prices once the seasonal climate is realized, evaluated before the seasonal outlook is released. Stdev_post is evaluated after the outlook's release and, in this example, is the same for either possible forecast. Option prices will reflect the decline from Stdev_pre to Stdev_post.

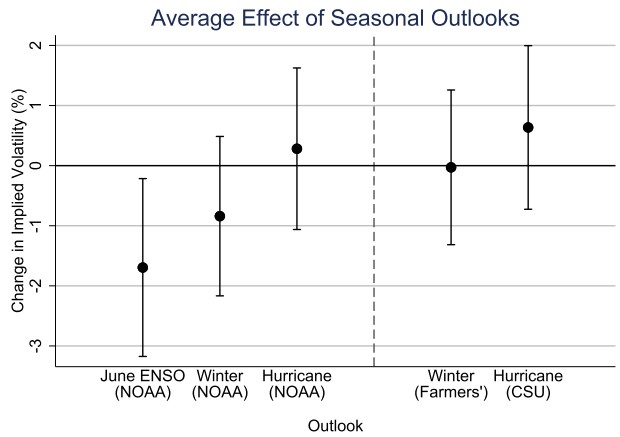

(a)  Estimated Effects

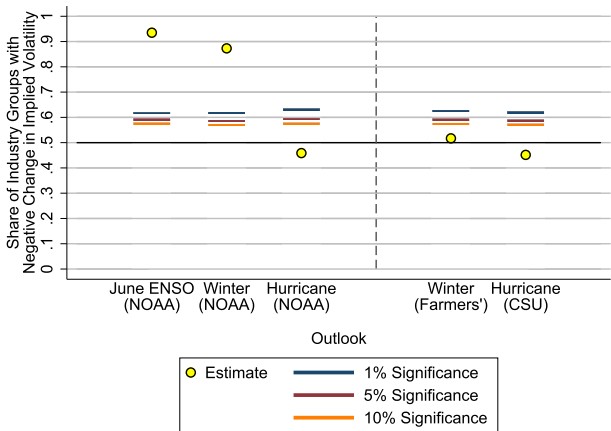

(b)  Share of Industry Groups with Effects

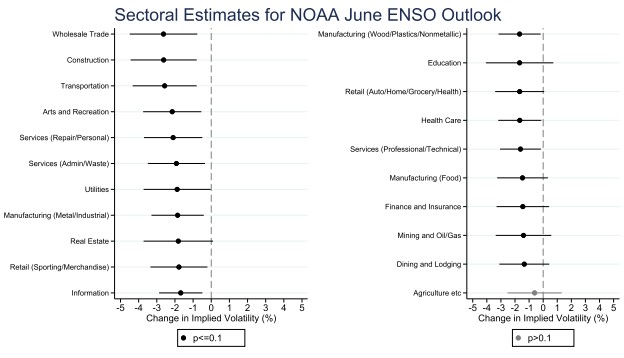

(c)  Sectoral Effects of June ENSO Outlook

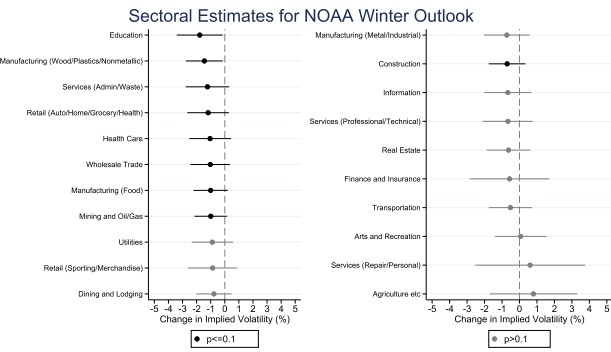

(d)  Sectoral Effects of Winter Outlook

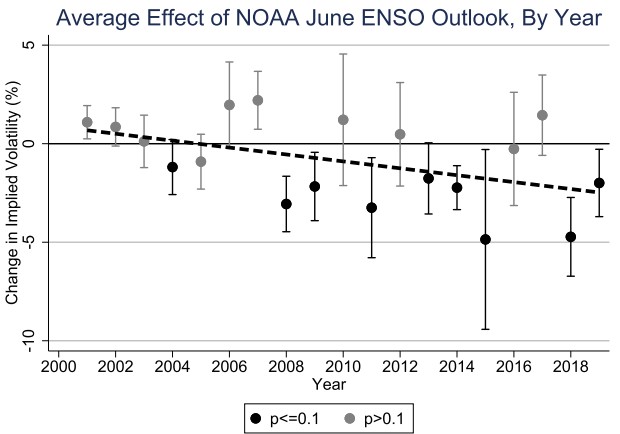

(e)  Effects by Year for the June ENSO Outlook

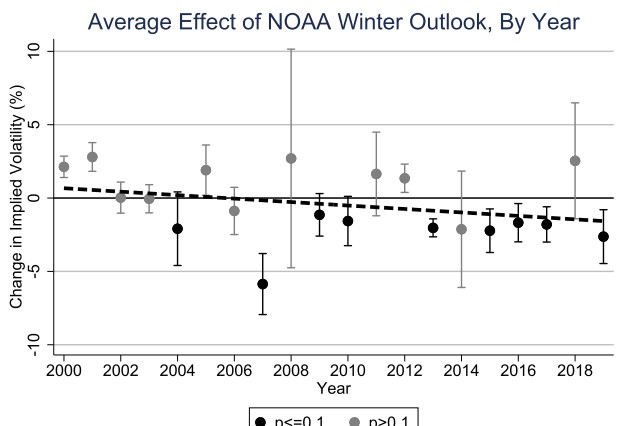

(f)  Effects by Year for the Winter Outlook

**Fig. 2 | Estimated effects of seasonal outlooks. a** Estimated average effects on implied volatility across all firms in 2010–2019, with 95% confidence intervals. **b** The share of industry groups for which the estimated effect on implied volatility is negative. Significance levels test the null hypothesis of an equal chance of positive or negative coefficients. **c**, **d** Estimated effects of the June El Niño Southern Oscillation and Winter Outlooks by sector, ordered by point estimates and with 95%

confidence intervals. **e**, **f** Estimated effects of the June El Niño Southern Oscillation and Winter Outlooks by release year, with 95% confidence intervals. The dashed line is the trend line across all years' estimates. **c–f** black markers indicate that the estimate is significant at the 10% level. Source data are provided as a Source Data file.

markets (see Methods). Negative estimates indicate that options traders did price uncertainty about outlooks' contents. The central estimates for both the June ENSO Outlook and the Winter Outlook are solidly negative. We can reject the null hypothesis of a weakly positive effect at the 5% level ($p$ value ($p$) = 0.013) for the June ENSO Outlook and nearly at the 10% level ($p$ = 0.11) for the Winter Outlook. In contrast, the Hurricane Outlook's central estimate is not significantly different from zero by any conventional measure, which could reflect traders judging the Hurricane Outlook to be less skillful or to be forecasting climate variables that are less relevant to stock market values. Supplementary Discussion 5 provides results in tabular form (Supplementary Table 4), reports several robustness checks (Supplementary Figs. 4–7), and shows that the strength of the June ENSO effect is a clear outlier within a placebo test of fake outlook release dates (Supplementary Fig. 8).

Figure 2b shows that the negative central estimates are not driven by a handful of outlier firms. It estimates average effects by industry group (defined by 4-digit North American Industry Classification System (NAICS) code, see Methods) and plots the share of industry groups with negative estimates. (The results in Fig. 2a implicitly account for the intensity of industry groups' effects.) We would expect ~50% of industry groups to have a negative estimate by chance (as seen for the Hurricane Outlook), but average implied volatility falls in ~90% of industry groups upon NOAA's release of either the June ENSO or Winter Outlook. These broad responses could reflect broad direct effects of forecasted seasonal climate or could reflect general equilibrium linkages that propagate effects on particular industry groups throughout the economy. Previous work has shown that the state of ENSO is important for the global economy[35–40] and that at least some producers do respond to ENSO outlooks[41]. Previous work has also shown that the prices of commodities (such as foods and metals) respond to the state of ENSO[42]. Such commodity market responses could represent supply-side effects that drive the broad industry exposure measured here or could instead represent demand-side effects that are driven by the broad industry exposure measured here.

Figure 2c, d reports estimates at the sector level (defined by 2-digit NAICS, see Methods), which aggregates similar industry groups. As the foregoing results suggest, nearly all sectors have negative point estimates. Moreover, of the 21 sectors, 20 are significant at the 10% level in the case of the June ENSO Outlook, and 9 are significant at the 10% level in the case of the Winter Outlook. We would expect to detect only two such sectors per outlook by chance. (Indeed, Supplementary Fig. 1 shows that the NOAA Hurricane Outlook significantly affects only one sector and misses the 10% cutoff by decimal places in another.) These differences could reflect, among other differences, the global nature of ENSO impacts as opposed to the U.S. focus of the Winter Outlook.

Some of the most affected sectors are intuitively exposed to the forecasted seasonal climate: the June ENSO Outlook significantly affects construction, transportation, and utilities, and the Winter Outlook significantly affects retail, health care, construction, and resource extraction. But many are less obvious, such as the strong responses of manufacturing and education. Moreover, the least-affected sector is, in each case, agriculture, which some might have expected to be especially sensitive to seasonal climate (although perhaps not to the winter climate targeted by the June ENSO and Winter Outlooks). These results imply that it would be a mistake to evaluate outlooks' effects by examining the difference in responses between sectors that an analyst judges ex-ante to be potentially exposed or not exposed to the seasonal climate.

We also analyze two non-NOAA outlooks in Fig. 2a, b: the Farmers' Almanac winter outlook (released in August) and the Colorado State University hurricane outlook (released in April). These outlooks garner substantial media attention, but prior literature suggests they are less

skillful[43,44]. If traders believe these outlooks to be as skillful as the NOAA outlooks, then we should find relatively stronger effects for the two non-NOAA outlooks because they are released before the associated NOAA outlooks and thus contain more novel information. However, if traders judge these outlooks to lack skill and do not reward media attention, then we should expect null results for the two non-NOAA outlooks. Our results fail to reject the null hypothesis of a weakly positive effect for either non-NOAA outlook. The estimated change in implied volatility is close to zero, and ~50% of industry groups show negative effects, as one would expect by chance. Supplementary Fig. 1 shows that neither outlook significantly affects even a single sector at the 10% level. Our results do not imply that markets never respond to information contained in particular releases of the two non-NOAA outlooks, but they do imply that markets do not expect these outlooks to contain relevant information on average. These comparisons suggest that the detected effects for the NOAA June ENSO and Winter Outlooks are real effects and also that markets appreciate skill, not just media attention.

Thus far, the analysis pooled years from 2010–2019 because experts in NOAA suggested that the quality of their outlooks improved around 2010 (see Supplementary Discussion 2). Figure 2e, f estimates effects by individual years and extend the analysis back to 2000. In contrast to the main analysis (see discussion of identification in Supplementary Discussion 4), each individual estimate is now vulnerable to chance events that happen on the day the outlook is released, so the reader should focus on trends across multiple years' estimates. The June ENSO and Winter Outlooks both appear to affect financial markets more strongly over time. For each outlook, only one year's estimate is significant between 2000 and 2006, whereas eight years' estimates are significant between 2007 and 2019. Each outlook's trend line slopes down, and Supplementary Fig. 3 shows that repeating the analysis of Fig. 2a over 2000–09 fails to detect significant effects. Supplementary Fig. 2 also shows that the other outlooks do not demonstrate a trend toward greater significance in later years, suggesting that the trend is not due purely to media attention. Instead, three other types of changes seem more likely to explain the increasing importance of seasonal outlooks to financial markets over time. First, predictive skill has indeed improved over time, most notably following the aforementioned upgrades to NOAA's forecast system around 2010[31,43,45,46]. Second, the outlooks have become more standardized over time, which may have increased their informativeness. Third, as the outlooks matured, process improvements may have reduced news leaking to markets ahead of time (see Supplementary Discussion 2). Of course, we cannot rule out that the changes over time were due to differences in market environments rather than to features of the outlooks themselves.

## Market value of seasonal outlooks

We use our estimated effects to measure the value of NOAA seasonal outlooks to financial markets (see Methods). Figure 3a measures the aggregate market capitalization of firms for which seasonal outlooks' effects are significant at various levels. At a 10% significance level, the June ENSO Outlook and Winter Outlook affect firms worth $13.4 trillion and $5.7 trillion, respectively, or 40% and 17% of our sample's market capitalization (equities that have options and survive our sample restrictions constitute around 75% of the total market's capitalization, Supplementary Table 5). At a 5% significance level, the June ENSO Outlook affects firms worth $9.2 trillion, and the Winter Outlook affects firms worth $4.1 trillion (28% and 12% of our sample's market capitalization, respectively). As not all of a firm's value will be exposed to the seasonal climate, these numbers are loose upper bounds on the market capitalization exposed to each seasonal outlook. The NOAA Hurricane Outlook and the two non-NOAA outlooks each affect firms with slightly less market capitalization than found for the NOAA Winter Outlook.

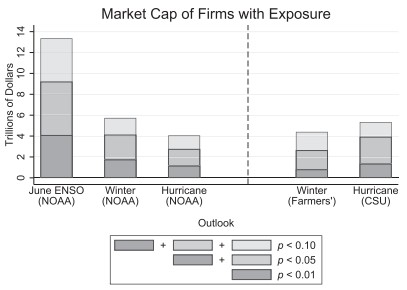

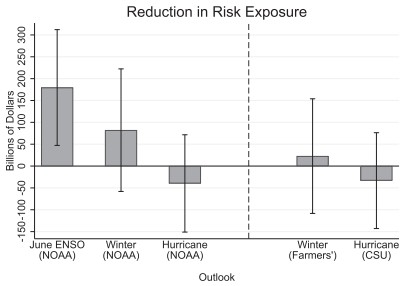

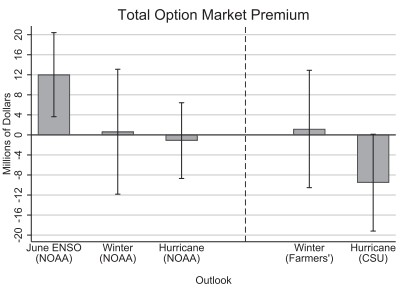

(a) Market Cap of Firms with Expo-  (b) Reduction in Risk Exposure  (c) Total Premium

sure

**Fig. 3 | Measures of aggregate value from seasonal outlooks. a** Market capitalization of firms for which estimated effects are significant at various levels; **b** reduction in market capitalization exposed to a one standard deviation risk; **c** option market premium induced by an upcoming seasonal outlook. Error bars indicate 95% confidence intervals. Source data are provided as a Source Data file.

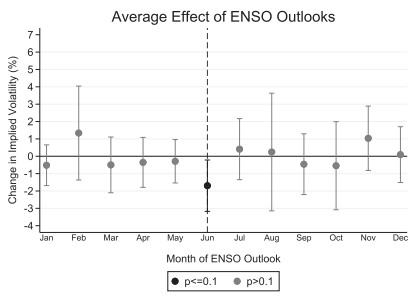

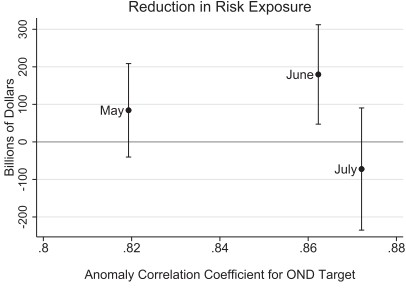

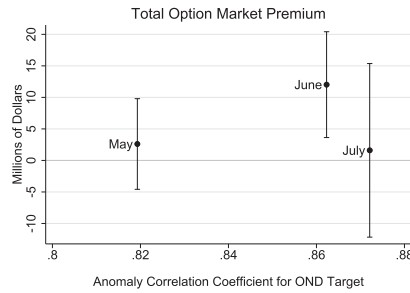

(a) Effects of ENSO Outlooks  (b) Reduction in Risk Exposure  (c) Total Premium

**Fig. 4 | Valuing the increase in skill from the May to June El Niño Southern Oscillation (ENSO) Outlooks. a** Estimated average effects of each month's ENSO Outlook on implied volatility, across all firms in 2010–2019. Black markers indicate that the estimate is significant at the 10% level. **b** Reduction in market capitalization exposed to a one standard deviation risk against the skill of the May, June, and July ENSO Outlooks. **c** Option market premium against the skill of the May, June, and July ENSO Outlooks. Skill is measured as the anomaly correlation coefficient for an October–November–December target. In all panels, error bars indicate 95% confidence intervals. Source data are provided as a Source Data file.

We also estimate the reduction in risk exposure from seasonal outlooks. Fig. 3b treats a firm's implied volatility as indicating the size of a one standard deviation risk and calculates the average reduction in this one standard deviation risk from releasing a seasonal outlook. The June ENSO Outlook and Winter Outlook reduce the market cap exposed to a one standard deviation risk by $180 billion [95% confidence interval (CI): $47–$312 billion] and $82 billion [-$58–$222 billion], respectively. The effect of the June ENSO Outlook is significant at the 1% level ($p = 0.0039$), and the effect of the Winter Outlook just misses significance at the 10% level ($p = 0.13$).

Figure 3c measures the total premium induced in options markets by the anticipation of an upcoming seasonal outlook. For the June ENSO Outlook, this premium amounts to $12 million [95% CI: $3.6–$20 million] annually. The premium for the June ENSO Outlook is significant at the 1% level ($p = 0.0025$). Because traders are willing to pay this premium when buying options, it can be interpreted as money spent hedging the risk of what the seasonal outlook might say.

Finally, we use the monthly schedule of ENSO outlook releases to value an increase in skill. As a rough estimate, we value improved skill by examining the change from the May to the June outlooks, which have similar lead times with respect to the boreal winter peak of ENSO. Measuring the ENSO Outlook's skill by its anomaly correlation coefficient for an October–December target (from ref. 47, see Methods), its skill increases by 5.2% from May to June. This substantial jump in skill

due to crossing the "spring barrier" is why we have thus far focused on the June outlook.

Figure 4a estimates the average effects on implied volatility from each month's ENSO outlook. Consistent with the one-off jump in skill upon moving past the spring barrier, the June Outlook is the only release that shows significant effects at the 10% level (and it is significant at even the 5% level, $p = 0.013$). In addition, Supplementary Fig. 9 shows that the June Outlook is one of only two outlooks (along with the May outlook) with a reduction in risk exposure that is significant at the 10% level (and it is significant at even the 1% level, $p = 0.0039$) and one of only two outlooks (along with the January outlook) with an option market premium that is significant at the 10% level (and it is again significant at even the 1% level, $p = 0.0025$). We would expect that 1 out of 12 tests would show significance at the 10% level purely by chance in each of these analyses, but the consistently significant result for the June outlook is precisely the one that we predicted would occur if traders are sensitive to the well-known increase in skill around the spring barrier.

Figure 4b shows that the June outlook reduces assets exposed to a one standard deviation shock by an additional $95 billion [95% CI: -$86–$277 billion] relative to the May outlook, implying that a 1% improvement in ENSO prediction skill reduces exposure by an additional $18 billion [-$16–$53 billion]. Figure 4c shows that the more skillful June outlook carries an option market premium that is $9.4

million [-$1.6–$20.5 million] larger than the May outlook, implying that a 1% improvement in ENSO prediction skill induces traders to spend an additional $1.8 million [-$0.31–$3.9 million] annually hedging news about seasonal climate. Figure 4c also shows that traders are not willing to pay much to hedge the news in the July Outlook, which is only marginally more skillful than the June Outlook and thus contains little new news. Supplementary Discussion 6 (along with Supplementary Table 6 and Supplementary Figs. 10, 11) shows that we estimate 30–40% larger effects of a 1% improvement in skill if we measure skill at forecasting either a November–January or December–February target instead of an October–December target.

## Discussion

Meteorological agencies currently prioritize improvements in seasonal forecasting[2,3], but it has been an open question whether seasonal forecasts are currently useful to the private sector. We measure the value of publicly funded seasonal outlooks to financial market participants. Our research design shows that financial options markets do price uncertainty about the contents of upcoming ENSO and winter seasonal outlooks: traders must believe that these seasonal outlooks may contain information relevant to firms' future performance. Importantly, our measures are lower bounds on outlooks' total value to society, as we do not measure nonmarket benefits, we ignore firms lacking liquidly traded options, and we do not measure the losses avoided by adaptation that uses outlooks' contents. Moreover, we measure only the incremental effect of seasonal outlooks against background information: some of the information in a given outlook will already be available through non-NOAA forecasting efforts, agents may extrapolate recent months' ENSO forecasts to generate current forecasts of ENSO, hurricane activity, or winter weather, and some of the contents of the Winter Outlook are used in the Energy Information Administration's Winter Fuels Outlook that is often released 1–2 weeks earlier. The total value of advance information about seasonal climate should include the value of this background information.

The fact that we detect sizable effects despite these limitations suggests that advanced information about winter and ENSO seasonal climate is rather valuable. Future work should investigate how other types of knowledge are transmitted from scientific communities to markets in order to understand the critical links. The fact that we do not detect such effects from the Hurricane Outlook suggests that this outlook may need to become more skillful and/or more specific if it is to provide value over and above shorter-run forecasts of particular storms. In fact, there are scientific efforts towards building future seasonal outlooks of U.S. land-falling hurricanes instead of the current operational basin-wide numbers (which may hit the U.S. or not)[48]. As hurricane outlooks advance in skill or are upgraded to include regional landfall, their impact on markets may change as it becomes easier to translate physical climate information (such as the number of storms) to financial impact.

Beyond the utility of seasonal outlooks, it has also been an open question whether seasonal climate patterns (as opposed to particular weather events in particular places) matter to the private sector. Such patterns are difficult to analyze because they affect weather in a variety of ways and over multimonth timescales. The release of a seasonal outlook collapses all these dimensions and timescales of realized weather into a discrete change in information on a particular day. Our results imply that seasonal climate patterns do matter–and they matter for firms throughout the economy. Future work should expand the economic analysis of weather beyond localized events to consider the import of large-scale atmospheric patterns.

Our results also have implications for the analysis of climate change impacts. First, financial regulators are grappling with the pricing of long-run climate risk[49–53]. A large and rapidly growing literature estimates the effects of weather events and of proxies for climate risk on stock prices[54,55]. We show that financial markets do price climate

risk at seasonal timescales. Moreover, traders expect the seasonal climate to affect firms throughout the economy, not just firms in sectors that are obviously exposed to the weather. This broad exposure to seasonal climate risk suggests the potential for broad exposure to long-run climate risks.

Second, our results suggest that adaptation to long-run climate change may face limits. Much work in economics projects the cost of climate change by extrapolating from the estimated effects of short-run weather anomalies[56,57]. However, that extrapolation presupposes that firms and citizens adapt to long-run climate change and short-run weather shocks in similar ways, which is an oft-questioned assumption[58–62]. Seasonal outlooks are intended to give agents and firms time to adapt to seasonal climates. If they could adapt costlessly and completely, then they would eliminate exposure to the forecastable component of seasonal climate risk, and investors would not need to hedge the contents of upcoming seasonal outlooks. Yet we do, in fact, infer that investors hedge the Winter and ENSO Outlooks. Therefore adaptation based on these outlooks must be incomplete and/or costly: firms are exposed to the seasonal climate despite the early warning, and/or firms do reduce their exposure but only at some nontrivial cost that affects their value on the stock market. These results complement other recent empirical work[62–65] in cautioning against expecting long-run adaptation to trivialize the costs of climate change. If firms cannot cheaply adapt to forecasted seasonal climate risks, they might also be unable to cheaply adapt to projected climate change risks.

## Methods

### Experimental design

We use event studies to detect whether option prices incorporate information about seasonal outlooks. Event studies isolate movements in financial variables due to the news released on particular days. They test whether the news released on days of interest is unusual relative to the news released on other days, after removing types of news that are explained by the controls. In our case, the event study removes news related to factors such as interest rates and earnings reports and then tests whether the remaining news on the days a seasonal outlook is released is unusual relative to other days in the sample.

In particular, we test the hypothesis that releasing an outlook reduces uncertainty. However, we cannot test this hypothesis by examining changes in option prices: the specific contents of an outlook release also affect option prices by changing the price of the underlying stock, and as a result of this effect and of changes in price due to changing time to expiration, the average change in option prices should be zero (under the risk-neutral measure, by familiar no-arbitrage arguments). We therefore test our hypothesis by testing whether the release of an outlook reduces options' implied volatility. Analyzing implied volatility is, in effect, a nonlinear way of controlling for the effects of changes in the stock price and time to expiration. Our specific methodological approach follows previous literature in running an event study in implied volatility[66–73].

As detailed in Supplementary Discussion 4, the effect of seasonal outlooks will be identified as long as other news that affects uncertainty about future stock prices is not systematically paired with the release of seasonal outlooks over the course of the decade. Of course, there will always be other news on any given day. We just need the news paired with outlook releases over the years to be as-good-as-random, net of our many controls. Supplementary Table 3 shows that obvious candidates for such news are not likely to be a problem, and robustness checks in Supplementary Discussion 5 show that our results are not sensitive to statistical specification.

### Financial data

Equity options data are from OptionMetrics. We use all firms available in IvyDB US (accessed through Wharton Research Data Services). This

broad sample biases us towards finding no effect because many of these firms may not operate in the geographical regions of interest to the seasonal outlooks studied here. The data include implied volatility calculations based on the binomial tree model[74]. This pricing model can be seen as a generalization (and discretization) of the Black-Scholes pricing model to allow for early exercise and account for dividends. As such, it does not account for "fat-tailed" risks or jump processes. We obtain firms' stock prices, quarterly dividends, earnings dates, and North American Industry Classification System (NAICS) classifications from Compustat.

We use the out-of-the-money call option that is nearest-to-the-money on a given day because this option's pricing tends to follow model predictions relatively closely[66,70,71]. We use the shortest-maturity options that expire at least a week past the end of our estimation window[75,76]. Forward-looking stock prices should internalize an outlook's news as soon as it is released, so options should reflect forecast-induced uncertainty regardless of whether they expire before or during the forecasted seasonal climate. The shortest-maturity options have two advantages: they tend to be more liquid, and they provide more power to detect an effect because the volatility induced by an outlook's release mechanically constitutes a larger share of volatility over the lifetime of the option when that lifetime is shorter[71]. We use only the main expiration date in a month (i.e., the third Friday of the month). Supplementary Discussion 3 describes additional sample restrictions imposed to ensure that we focus on liquidly traded options. Supplementary Fig. 5 reports results for longer-maturity options (which have liquid options for only half as many firm-years). Supplementary Fig. 6 reports the results for put options.

## Statistical analysis

Let $t$ indicate trading days, $i$ indicate firms, $f$ indicate forecasts (i.e., seasonal outlooks), $y$ indicate years, and $m$ indicate the market, which we will define below. We estimate the following regressions:

$$\ln\left(IV_{it}/IV_{i(t-1)}\right) = \sum_{j=-1}^{1} \beta_{fmj}D_{f(t+j)} + \Gamma_{iy}X_{it} + \epsilon_{it}, \quad (1)$$

where $IV$ is implied volatility (described above) and $D_{fs}$ is a dummy variable that equals one if forecast $f$ is released on day $s$ and is zero otherwise. $X_{it}$ is a vector of controls, with $\Gamma_{iy}$ a coefficient vector that allows the effects of the controls to vary by firm and by year. In the authors' preferred specification, the controls include a constant (which implies firm-year fixed effects), the option's time to expiration and its square, the log-change in the London Inter-Bank Offered Rate (LIBOR, which was a benchmark interest rate tied to prominent banks' willingness to lend to each other), the log-change in the 10-year Treasury rate, and dummy variables for a 3-day event window centered on any earnings announcements for firm $i$ (we drop any firm-outlook-year triplet with an earnings report in the 3-day event window). Supplementary Table 4 reports measures of sample size.

The preferred specification's controls help absorb any tendency of implied volatility to change over time, absorb any persistent pricing model errors in the implied volatility calculations, and absorb the effects of any news not directly related to the seasonal outlooks, without including controls (such as the S&P 500) that would absorb the effects of outlooks that move many firms.

Each regression in our main analysis uses multiple years' outlook releases. Within each year, we use a 30-day estimation window centered around a 3-day event window. The 30-day estimation window means that we compare changes in implied volatility on the day an outlook is released to changes on nearby days, within the same season. The 3-day event window effectively removes the day before ($j = 1$) and day after ($j = -1$) the forecast release from the sample. In order to favor more liquid observations, we weight observations by the inverse of the relative bid-ask spread[77,78].

We are interested in $\beta_{fm0}$. Supplementary Discussion 4 discusses its identification. This coefficient tells us how the implied volatilities of a market's options change, on average, on the day that an outlook is released. The implied volatility on day $t$ represents the average of the daily volatilities expected over the remaining life of the option[79,80]. Finding that $\beta_{fm0} < 0$ in some given year could reflect that publishing seasonal outlooks means (1) agents no longer face volatility induced by uncertainty about the contents of the outlook and/or (2) agents learn from the outlook that the seasonal climate will be especially stable. We want to isolate the first effect. Rational expectations imply that the second effect should be zero on average. Averaging many years' outlook releases should thus isolate the first effect[70], as derived in Supplementary Discussion 1. We will detect an effect only if investors believe that the outlooks are skillful *and* believe that market $m$ is sensitive to weather in the forecasted season.

We conduct the analysis at various levels of aggregation. First, when we estimate a single coefficient per outlook, we pool all firms together and estimate a common effect across them (so that $m$ indicates the whole market, which is thousands of firms). Second, when we assess the fraction of industry groups with negative central estimates, we aggregate to the 4-digit NAICS codes (so that $m$ indicates a 4-digit code). We include only industry groups with at least five firms. Third, when we plot effects by sector, we aggregate to 2-digit NAICS codes (so that $m$ indicates a 2-digit code). We include only sectors with at least 5 firms. When aggregating above the firm, we cluster standard errors by date to account for potential correlation across firms. We use heteroskedasticity-robust standard errors when running regressions at the firm level. Reported p-values reflect a null hypothesis of a weakly positive effect (i.e., we use one-tailed tests with the critical region in the left tail). The significance cutoffs for the share of negative industry group coefficients come from the binomial distribution, reflecting a null hypothesis that positive and negative coefficients are equally likely.

Results are robust to dropping the controls and the weights and to additionally clustering by firm-year (see Supplementary Figs. 4 and 7). One may consider using options on the S&P 500 index instead of options on individual firms. The problem is that the number of observations falls from hundreds of thousands to ten. Tests with the S&P 500 index do not clearly contradict the results reported in the paper, but in contrast to the paper's results, the central estimates do become rather sensitive to choices such as weighting schemes and controls.

## Value calculations

For all value calculations, we drop all assets with NAICS code 5259 ("Other Investment Pools and Funds"). These are mostly exchange-traded funds, or ETFs. They can be large, their estimates can be sensitive to specification, and they are not the types of equities of interest here.

We estimate a version of equation (1) modified to permit the effects of a forecast release to vary by firm:

$$\ln\left(IV_{it}/IV_{i(t-1)}\right) = \beta_{fi0}D_{ft} + \beta_{f(-1)}D_{f(t-1)} + \beta_{f1}D_{f(t+1)} + \Gamma_{iy}X_{it} + \epsilon_{it}.$$

We use the estimated $\hat{\beta}_{fi0}$ as described below. The point estimates resulting from the calculations described below are extremely similar if we instead obtain the $\hat{\beta}_{fi0}$ by estimating equation (1) firm by firm.

Consider the reduction in risk exposure. For each seasonal outlook $f$, we calculate

$$\sum_i \left\{ \left[1 - e^{\hat{\beta}_{fi0}}\right] IV_{i(\tau-1)} MarketCap_i \right\},$$

where day $\tau$ is the outlook release date, $IV_{i(\tau-1)}$ is the average implied volatility across the relevant expiration date's strikes weighted by open

interest, and $MarketCap_i$ is the firm's market capitalization at the end of the most recent fiscal year prior to 2020 that has available data (this is typically 2019), from Compustat Fundamentals Annual. (Approximately 85 percent of firms end their fiscal year on December 31.) The exponentiation and multiplication by $IV_{i(\tau-1)}$ give the reduction in implied volatility due to the release of an outlook, adjusting for the regression having used the log-change in IV as the dependent variable. Multiplying by market capitalization gives the total value of stock exposed to the reduction in a one standard deviation risk. We obtain the standard error for each forecast by the delta method, using the full covariance matrix of the $\hat{\beta}_{fi0}$.

Now consider the total option premium. For each outlook $f$, we calculate

$$\sum_i \left\{ \left[ e^{-\hat{\beta}_{fi0}} - 1 \right] \sum_e \sum_K IV_{iKe\tau}\, \nu_{iKe\tau}\, OpenInterest_{iKe(\tau-1)} \right\},$$

where day $\tau$ is the outlook release date, $K$ indexes strike prices, $e$ is the expiration date, and $\nu$ is the option's vega (i.e., its sensitivity to implied volatility, as calculated by OptionMetrics). The exponentiation and multiplication by $IV_{iKe\tau}$ give the increase in implied volatility due to an upcoming outlook, adjusting for the regression using the log-change in IV as the dependent variable. Multiplying by vega converts this measure to a change in the option price, and multiplying by open interest then gives the change in value for all traded options. We use the implied volatility and vega from the 2019 outlook release date, and we use date $\tau$ (and include a negative sign in the exponentiation) because we want to predict implied volatility leading up to the outlook release, not after the outlook is released. Open interest is for the day before the outlook release from the last year (through 2019) for which the firm appears in the data. We obtain the standard error for each forecast by the delta method, using the full covariance matrix of the $\hat{\beta}_{fi0}$.

We measure the ENSO Outlook's skill by the anomaly correlation coefficient for forecasts of a 3-month target. Based on expert advice, we use an October–December target as that window is especially informative about boreal winter conditions. We choose not to regress each monthly outlook's value measures against its skill because differences in lead times could directly affect value even if skill were constant. Instead, we focus on the May, June, and July outlooks because the change in lead time is minor from June to either adjacent month and skill is known to jump between the May and June outlooks. Using author-provided tabular data from ref. 47, the anomaly correlation coefficient over 1982–2010 is 0.8193 for the May Outlook, 0.8623 for the June Outlook, and 0.8721 for the July Outlook. We calculate the value of a 1% improvement in outlook skill by linearly extrapolating the estimated effect from the actual May-to-June change in skill to the effect of a 1% improvement from the May outlook's skill. The standard error derives from the square root of the summed squared standard errors of the May and June option market estimates. Supplementary Discussion 6 (along with Supplementary Table 6 and Supplementary Figs. 10 and 11) assesses sensitivity to other forecast targets.

### Reporting summary
Further information on research design is available in the Nature Portfolio Reporting Summary linked to this article.

## Data availability
The raw option price data are restricted per license from Option-Metrics. Access to intermediate data files can be obtained with permission from OptionMetrics. The data generated in this study via analysis of options data have been deposited in the ICPSR database as project openicpsr-199192 with available at https://doi.org/10.3886/E199192V1[81]. Source data are provided with this paper.

## Code availability
All code is Stata code for cleaning data, running regressions, performing calculations, and plotting results. All analysis uses publicly available functions, not custom algorithms. The code created for this study has been deposited in the ICPSR database as project openicpsr-199192 with available at https://doi.org/10.3886/E199192V1[81].

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

## Acknowledgements
Austin Drukker and Paul Fisher provided research assistance, and Robin Sehler, Frances Slater, and Ryan Ng collected data. We thank Monica Grasso, Dave DeWitt, Matthew Rosencrans, Michelle L'Heureux, and Hiroyuki Murakami for their comments and Catherine Raphael for graphical design support. We are grateful for funding from NOAA's Climate Program Office (CSI SARP, award NA18OAR4310261, to DL as PI and SK as co-PI) and to NOAA for support for Robin Sehler (under award NA18OAR4320124, to SK). The statements, findings, conclusions, and recommendations are those of the author(s) and do not necessarily reflect the views of NOAA or the U.S. Department of Commerce. The University of Arizona's High Performance Computing (HPC) facility provided an allocation of computer time.

## Author contributions
DL and SK developed the idea and designed the experiments. SK collected the seasonal outlook data. DL organized the financial data and designed and implemented the statistical analysis.

## Competing interests
There are no competing interests to declare for DL. SK conducted the research while employed and funded at NOAA without competing interests but subsequently worked for some time in climate strategy at a bank. To avoid conflict, financial data and statistical analysis reside with DL without access for SK.
