## [Peer review file · Nature Communications]

Reviewers' comments:

Reviewer #1 (Remarks to the Author):

This is an important study in the field of climate change focusing on evaluating the scientific research in forecast. However, there are a couple fundamental problems preventing accept the publication in the current form

Major question

1. The key theory for this study is to assume that releasing outlook will reduce the uncertainty by the law of total variance. However, there two possible ways driving it not appropriate: First, the law of total variance is saying that all possible outcomes are known even we do get the possibility before the release of outlook. However, some volatile climate sometimes have not been known until now which is the direct undetermined factor to increase the total variance. Once this kind of new knowledge which is not known or understood by traders, the uncertainty of stock price may not decrease; Second, W_t assumed correctly measured, but it is not necessarily 100% correct. All mistakes in the outlook could increase total variance too. As these possible theoretical problems, we recommend authors remove the model, and argue findings are limited.
2. The whole article try to argue that NOAA outlooks have more skill than the report from other two outlooks. However, it is not quite clear to see it is true or not. Can authors provide evidences for comparison?
3. How to measure implied volatility? It is not obvious for audience to understand.

Minor questions

1. In Figure 2, why hurricane outlook has much smaller impact on volatility? Usually the damage driven by hurricane is much sever than ENSO, isn't it?
2. In figure 2, why the effect in 2007 is much negative than others?
3. On line 74, p-value is 0.11, we cannot claim it is significant at 10%. Actually, it is not significantly differeent from 0.
4. The share of industry groups impacts is not informative as the information from outlook is not equally valuable for different industries. Typically, it is related to the sample, and consider those differences have been captured in the next figure, I would recommend to drop it.
5. It is confused to state "effects of each month's ENSO Outlook", we only know ENSO outlook is releasd in June. Once the information is released, all values will be in prices.

6. Did authors control for temporal variables such as weekdays as we find release days are not always the same?

Reviewer #2 (Remarks to the Author):

The aim of this article is to study the value financial markets attach to longer-term weather forecasts (mainly the ENSO outlook by NOAA).

The analysis seems competently conducted (with the caveat I have not tried to replicate the results, nor I have gone through the technicalities in detail). It reads as a solid piece of work from an empirical and financial perspective. In principle, I would be in favour of seeing this published in Nature Communications.

However, I would like to suggest two main things to the authors.

First, I think the writing style could be improved by making things clearer, especially to the reader unexperienced with finance. Certain important bits of info are relegated to the Methods section: the most prominent example is the very first paragraph of Methods, where (finally!) authors offer readers the clear synthetic info of what they're talking about. This is not 'methods', but rather essential info the reader needs to read to understand from the very start so to appreciate what comes next. I was rather confused on some details explained only later in the text (e.g. which outlooks are you analyzing; or what do you mean by 'skill'). If I'm not mistaken, even ENSO is never really explicitly defined (if one doesn't know what it is, why should they care?). I can see the authors already put an effort in explaining options and other concepts in clear terms, but I feel they are not there just yet.

Second, and probably even more important, I think the authors should present better why this analysis is relevant and interesting. Financial markets put some value in the uncertainty on ENSO and its resolution. So what? Why should I be interested in this? Is the value particularly high or low? Does this mean something special? While I appreciate the competence of the analysis, I am slightly struggling in understanding what to do with it. Haven't weather outlooks always been relevant for markets? I think the authors are trying it to connect the topic to climate risks in the discussion section, but I didn't find it compelling. I also struggle to perceive this study to be about climate change, as the authors seem to suggest. Is the study about the change in ENSO frequency or strength? If not, isn't the paper only about climate, per se? ENSO and hard winters have been happening also before anthropogenic change.

Referee 1

Thank you for your helpful comments and guidance. We do our best to answer all of your questions below.

Your comments are below, `boxed in with a gray background and different font`. My responses are in plain text. Any included text from the new manuscript is `boxed in with a white background and the same font as this text` with the section number noted at the top of the quoted text.

Referee 1: Major Comment 1

Referee 1: Comment 1

This is an important study in the field of climate change focusing on evaluating the scientific research in forecast.

We appreciate the kind evaluation.

Referee 1: Comment 1

However, there are a couple fundamental problems preventing accept the publication in the current form

Major question 1. The key theory for this study is to assume that releasing outlook will reduce the uncertainty by the law of total variance. However, there two possible ways driving it not appropriate: First, the law of total variance is saying that all possible outcomes are known even we do get the possibility before the release of outlook. However, some volatile climate sometimes have not been known until now which is the direct undetermined factor to increase the total variance. Once this kind of new knowledge which is not known or understood by traders, the uncertainty of stock price may not decrease; Second, W_t assumed correctly measured, but it is not necessarily 100% correct. All mistakes in the outlook could increase total variance too. As these possible theoretical problems, we recommend authors remove the model, and argue findings are limited.

Law of Total Variance The referee objects to our theoretical analysis in Section A of the SI. In that analysis, we use the law of total variance to decompose uncertainty about a future stock price into expected uncertainty once the seasonal outlook is known and the variance of expected stock prices upon observing a seasonal outlook (reflecting how the unknown contents of an upcoming seasonal outlook can affect the stock's value). Simple rearrangements yield the key prediction we seek to test. The beauty is that these are all straightforward mathematical relationships that hold for any pair of random variables (here, the future stock price and the upcoming outlook's contents). The relevant material from SI A is:

From SI A

Let T be the date an option expires and let $\tau < T$ be the date an outlook is released. The time t stock price is S_t . Let W_τ denote the contents of the outlook, which are known as of τ but not any earlier. From the law of total variance,

$$\text{Var}[S_T] = E[\text{Var}[S_T|W_\tau]] + \text{Var}[E[S_T|W_\tau]].$$

Rearranging,

$$E[\text{Var}[S_T|W_\tau]] - \text{Var}[S_T] = -\text{Var}[E[S_T|W_\tau]].$$

The first term on the left-hand side is the variance of S_T averaged over all the possible outlook contents W_τ , and the second term on the left-hand side is the variance of S_T without knowing the outlook contents W_τ . By iterated expectations, this becomes

$$E[\text{Var}[S_T|W_\tau] - \text{Var}[S_T]] = -\text{Var}[E[S_T|W_\tau]].$$

The left-hand side is the expected change in variance upon the release of an outlook. If we observe many outlook releases, the left-hand side gives the average change in variance upon the release of the outlook. The right-hand side is the negative of the variance in the stock price induced by the possible outlooks. The right-hand side is weakly negative, and it is strictly negative if the outlook's estimates might be relevant for the stock price. Therefore the variance of S_T should fall on average upon the release of an outlook if the outlook is potentially informative for stock prices, and the variance of S_T should not change on average otherwise. We test for the average decline in variance by using the change in the variance implied by options' prices.

The referee argues that we miss two factors important to the real world: (1) some types of volatile climate may be previously unknown (i.e., there could

be surprises), and (2) the outlook's prediction (our W_τ) could be mistaken. We are sure that neither of these possibilities poses a problem for our analysis. We will try our best to summarize what we think the referee's concerns are.

(1) First, we agree with the referee that we all learn about climate over time, especially in a world of climate change. Further, we do understand that weather events may happen that are extremely surprising, and we admit that there could be some chance that some weather events could be so surprising that they were not previously considered possible (and that such an event could increase uncertainty about later weather). However, we want to emphasize that our analysis describes the effects of releasing an outlook that encompasses weather events several months out; it does *not* describe the effects of particular weather events that happened to be realized. We cannot imagine that any weather event so rare as to have been considered impossible would ever be forecasted months ahead of time by a NOAA seasonal outlook, and this certainly did not happen in the years we study.

That is our informal reasoning. Attempting to formalize the referee's objection, we think the referee may be suggesting that the variance of stock prices after observing an outlook may underestimate the potential weather events that could occur: that $Var[S_T|W_\tau]$ is smaller than it would be if traders were cognizant of all weather possibilities. But such an effect causes no problem for us. Even if investors do underestimate the potential for weather surprises, we are pricing only whatever it is that traders do anticipate, not whatever weather happens to be realized down the line. All we are concerned with is investors' perceived variance. As such, our mathematical derivation is unaffected by the referee's objection.

It is possible that the referee is instead arguing that variance may not decline following some types of observations. If this is the argument, it is explicitly permitted by both our theoretical analysis and our empirical framework. Our formal result is that variance must decline on average, not in every instance. This is precisely why our empirical tests average over a decade of outlook releases. Indeed, we wrote in the main text, "Even if some particular outlook's release happened to increase traders' uncertainty by forecasting an especially volatile climate, the law of total variance implies

that releasing outlooks should reduce their uncertainty on average (see Supplementary Information A).”

In order to alleviate this first concern of the referee’s, we have amended the following paragraph in the introduction to make it clear that we refer to traders’ uncertainty (rather than “correct” beliefs) about the weather.

From Introduction

Instead, our methodology tests whether traders’ uncertainty about future stock prices tends to fall upon the release of an outlook. In Figure 1, the standard deviation of future stock prices is smaller once the outlook is released. Even if some particular outlook’s release happened to increase traders’ uncertainty by forecasting an especially volatile climate, the law of total variance implies that releasing outlooks should reduce their uncertainty on average (see Supplementary Information A). We measure traders’ uncertainty from the standard deviation implied by option prices, commonly referred to as “implied volatility” (see Methods). If markets anticipate outlook releases, then each affected firm’s implied volatility should, on average, decline when outlooks are released. We therefore increase our power to detect an effect in our base analysis by combining information from thousands of firms’ responses and from ten years of outlook releases.

And we have amended the mathematical derivation in SI A to also make it clear that the variance referred to is traders’ perceived variance.

(2) Second, we of course agree that the outlook’s prediction could be mistaken. These are seasonal outlooks! Even short-run weather forecasts are often mistaken. However, we definitely did not assume that the seasonal outlook cannot be mistaken. In fact, we did not even assume that it is any good. Our claim is that we are testing whether traders perceive it to be conveying any news at all, not whether that news is a perfect forecast of multimonth weather.

In terms of our formalism, we believe that the referee’s point is that $Var[S_T|W_\tau]$ (the variance of future stock prices conditional on knowing the

outlook's contents) should include the variance induced by the possibility of forecast mistakes. Fortunately, it definitely does! That variance includes any and all random variables realized between time τ (when the outlook is released) and time T (when the option expires). Such random variables include weather that was not forecasted and also weather that contradicts the seasonal outlook's forecasts. Again, these random variables are from the perspectives of traders at time τ . In the highly unlikely event that traders do ignore the possibility that seasonal outlooks may not produce perfectly correct forecasts, our analysis is still correct because, as we emphasize in the text, we are trying to measure whether traders perceive outlooks to have skill (even if traders' perceptions were somehow incorrect).

For completeness, one might wonder how our formalism would apply to a seasonal outlook that traders perceived to have zero skill. The release of such an outlook would not move stock prices at all, so that the variance of stock prices would always be the same whether conditioned on the outlook's contents or not:

$$Var[S_T|W_\tau] = Var[S_T]$$

for all W_τ . In that case,

$$E[Var[S_T|W_\tau]] = Var[S_T].$$

But this implies that the left-hand side of our key relationship in SI A is zero: we should not find any effect! Indeed, this possibility is part of the null hypothesis we test.

In order to alleviate this second concern of the referee's, we have amended our derivation to make it clear that our variance includes unforecasted weather:

From SI A

Either variance includes randomness in weather that might occur by time T , all other random shocks that might affect firm value by time T , and the possibility that outlooks' predictions are revised by information arriving before time T . However, only the second variance on the left-hand side includes uncertainty about what the outlook will say.

We have also clarified what it means if traders perceive a seasonal outlook to lack any skill:

From SI A

For completeness, one might wonder how our formalism would apply to a seasonal outlook that traders perceived to have no skill. The release of such an outlook would not move stock prices at all, so that $E[S_T|W_\tau]$ is independent of W_τ . In that case, $Var[E[S_T|W_\tau]] = 0$. So the average change in traders' variance upon the release of the outlook must also be 0: we should not find any effect of releasing an outlook. This possibility of no skill is included within the null hypothesis we test.

We thank the referee for pushing us to clarify. We have worked hard to charitably interpret the referee's concerns and apologize if we have misunderstood one of the objections. In that case, we ask that the referee clarify further so we can offer what assistance and edits we can. In the end, we believe that our use of the law of total variance is not affected by the referee's two concerns and hope that the referee agrees.

Referee 1: Major Comment 2

Referee 1: Comment 2

2. The whole article try to argue that NOAA outlooks have more skill than the report from other two outlooks. However, it is not quite clear to see it is true or not. Can authors provide evidences for comparison?

Skill of Outlooks We are testing whether traders perceive NOAA outlooks to have skill at predicting relevant weather. As we write,

From Introduction

Options markets should reflect the uncertainty induced by an upcoming seasonal outlook if its possible forecasts would affect firm value and thus stock prices. They should not reflect the uncertainty induced by an upcoming seasonal outlook if traders do not judge the outlook to be skillful, if traders judge seasonal climate to be irrelevant to profits, or if the outlooks' long lead times allow firms to cheaply minimize exposure to the forecasted seasonal climate.

The referee refers to our claims about the relative skill of outlooks. We make such claims only when we try to validate our results by testing two outlooks widely thought to lack skill. Those two outlooks are the Farmers' Almanac and Colorado State University outlooks. The former is not even based on a scientific model, so one would hope that the NOAA outlook is more skillful. And the latter was the original hurricane outlook but has been found by some to actually perform worse than one would do by chance. In these cases, one might expect to find a null effect if traders prioritize skill, and that is exactly what we find.

Conducting new tests of outlooks' actual skill would be a significant paper in its own right and well beyond the scope of the present paper. Fortunately, though, others have done such tests already. And we did cite that work to justify our claims. We previously included those citations only in the second

paragraph of the Methods. The other referee requested that we move the background material on seasonal outlooks into the main text. In doing so, we now provide the additional background and citations regarding NOAA seasonal outlooks upfront in the introduction:

From Introduction

We test whether options markets in 2010–2019 priced uncertainty about the news contained in upcoming seasonal climate outlooks for winter weather, the El Niño Southern Oscillation (ENSO), and hurricanes. The U.S. National Oceanic and Atmospheric Administration (NOAA) releases each seasonal outlook on a regular, announced schedule with strict rules to guard against information leaking in advance (see Supplementary Information B). Each product provides multi-month predictions (and hence is a “seasonal” outlook). “Skill” refers to the accuracy of an outlook’s forecasts over many years. The Atlantic Hurricane Seasonal Outlook is released in May and includes outlooks for both the Atlantic and Eastern Pacific basins. The U.S. Winter Outlook is typically released on the third Thursday of October. It reports the probability that each part of the country will experience abnormal seasonal temperatures or precipitation over the coming December through February. The ENSO seasonal outlook is released monthly. ENSO refers to sea surface temperature and wind anomalies in the eastern Pacific. The state of ENSO is often found to predict climate variables elsewhere in the world, including temperature and precipitation.²³ Each month’s outlook reports the current state of ENSO and provides predictions out to 9 months. The June ENSO Outlook is known to be especially informative because it is the first to take advantage of the jump in skill after the “spring barrier”.^{24–29} This jump in skill arises in part because contributing factors to ENSO are especially noisy in the spring.³⁰

We believe that citing this work in a more prominent location mitigates the referee’s concern. And even more directly to the referee’s concern, our main

text now includes the citations for non-NOAA outlooks that were previously only in the Methods:

From Results

We also analyze two non-NOAA outlooks in Figures 2a–b: the Farmers’ Almanac winter outlook (released in August) and the Colorado State University hurricane outlook (released in April). These outlooks garner substantial media attention but prior literature suggests they are less skillful.^{33,34}

We hope the referee agrees that the clearer organization of the revised manuscript allays their concern.

Referee 1: Major Comment 3

Referee 1: Comment 3

3. How to measure implied volatility? It is not obvious for audience to understand.

Measuring Implied Volatility Following much other work, we did not directly implement these calculations for our study. Instead, we used the benchmark implied volatility calculations that are used throughout the finance literature on equity options. In the Methods, we cite the source of the data and, going above and beyond most comparable work, we even provide a citation for the methods that the data provider uses to calculate implied volatility:

From Methods

Equity options data are from OptionMetrics. We use all firms available in IvyDB US (accessed through Wharton Research Data Services). This broad sample biases us towards finding no effect because many of these firms may not operate in the geographical regions of interest to the seasonal outlooks studied here. The data include implied volatility calculations based on the binomial tree model.⁶⁴

The methods are both technical and conventional enough that we are not going to be able to provide a complete description in the Methods. Fortunately, however, an interested reader can refer both to the OptionMetrics user guide and to the paper we cite, and any number of web sites provide introductions to binomial tree models. We hope that this additional explanation and the pointer to our Methods description alleviates the referee's concern.

Referee 1: Minor Comment 1

Referee 1: Comment 1

Minor questions 1. In Figure 2, why hurricane outlook has much smaller impact on volatility? Usually the damage driven by hurricane is much sever than ENSO, isn't it?

Hurricanes' small effects The referee wonders why we do not find larger effects for hurricanes. We want to emphasize that we are *not* testing which type of weather matters more for the U.S. economy. Instead, we are jointly testing which outlooks traders find to be skillful *and* which type of forecasted weather traders believe is important to the U.S. economy. Failing to find an effect of the hurricane outlook can reflect that traders do not perceive it to be skillful in the ways relevant to them, that traders are not concerned with the weather it predicts, or both.

The referee also wonders why the hurricane outlook does not have a stronger effect than the ENSO outlook. It should be clear by now that this difference could be due to skill, but let's for the moment assume that each outlook is equally good at forecasting whatever variables they do forecast. It may then be that the hurricane outlook forecasts variables that are not the ones that traders care about. In particular, each outlook is a basinwide forecast, but hurricanes primarily cause damages if they hit land, and even then whether they cause much damage depends on where exactly they will make landfall. It may be that the relatively coarse basinwide forecasts contained in current outlooks are simply not providing information of sufficient relevance to traders.

Now let's assume that each outlook is equally good at forecasting the types of variables traders find important. Even then it is not obvious that the hurricane outlook should matter more. We show that ENSO has effects throughout the U.S. economy, affecting virtually every sector. *If* a large hurricane happens to hit the right spot, it can inflict substantial localized damage, but it is far from obvious that even that worst-case outcome would have a stronger effect on the overall U.S. economy than does the systematic shift induced by ENSO across the whole U.S. over the whole fall/winter/spring. Only the very worst hurricanes damage property worth more than a tenth of a percent of GDP. Once we also account for a large strike to an important area being relatively unlikely even if the hurricane outlook were to forecast a good number of storms, it is not surprising that ENSO would matter more.

Last, we want to note that Figure S-1 reports sectoral estimates for the hurricane outlook. We do find one sector with a significant effect, but as we note there, this is precisely what one would expect by chance (and extremely different from what we found for the ENSO and Winter Outlooks in Figure 2c-d).

In order to address the referee's concern, we have elaborated on what our hurricane result means and why it may not be surprising. We have added the following to the Results:

From Results

In contrast, the Hurricane Outlook's central estimate is not significantly different from zero by any conventional measure, which could reflect traders judging the Hurricane Outlook to be less skillful or to be forecasting climate variables that are less relevant to stock market values.

And we have added the following paragraph to the Discussion:

From Discussion

The fact that we detect sizable effects despite these limitations suggests that advance information about winter and ENSO seasonal climate is rather valuable. And the fact that we do not detect such effects from the Hurricane Outlook suggests that this outlook may need to become more skillful and/or more specific if it is to provide value over and above shorter-run forecasts of particular storms. In fact, there are scientific efforts towards building future seasonal outlooks of U.S. land-falling hurricanes instead of the current operational basin-wide numbers (which may hit the U.S. or not).³⁸ As hurricane outlooks advance in skill or are upgraded to include regional landfall, their impact on markets may change as it becomes easier to translate physical climate information (such as number of storms) to financial impact.

We hope that these explanations satisfy the referee.

Referee 1: Minor Comment 2

Referee 1: Comment 2

2. In figure 2, why the effect in 2007 is much negative than others?

Interpreting Figure 2e–f The referee wonders why effects were so strong in 2007. We want to strongly caution against overinterpreting any one point in this figure. We intentionally used this figure only to explore trends and intentionally discussed significance only over groups of years. In our main analysis, each estimate relies on 10 years of outlook releases. In any given year, something random would have happened on the same day an outlook is released, but over many years, these types of effects cancel out, enabling us to estimate the average effects of outlooks (see SI Section D). By this same logic, trends over years in Figure 2e–f are interpretable as trends in the effects of outlooks, but any one year’s effect could capture the randomness of something else just happening to have happened on that same day. And because that something else could be just about anything, we do not want to go down the rabbit hole of trying to explain each single estimate, as there are countless stories one could tell. It is better to stick to what is statistically meaningful, which are the average effects over a decade and the trends over years.

Also, we should clarify that we do not predict that the release of a seasonal outlook will move variance the same way for every year—or even reduce it in every year. Our formal result is that variance must decline on average, not in every instance. This is one reason why our primary empirical tests average over a decade of outlook releases.

In order to clarify for other readers, we have added the following text to the main text:

From Results

In contrast to the main analysis (see discussion of identification in Supplementary Information D), each individual estimate is now vulnerable to chance events that happen on the day the outlook is released, so the reader should focus on trends across multiple years’ estimates.

We hope the referee agrees with our decision to refrain from commenting on any particular year’s estimate in Figure 2e–f and finds our additional explanation useful.

Referee 1: Minor Comment 3

Referee 1: Comment 3

3. On line 74, p-value is 0.11, we cannot claim it is significant at 10%. Actually, it is not significantly different from 0.

Significance of Winter Outlook effect The referee wants us (i) to avoid claiming significance at the 10% level and (ii) also to go further and claim non-significance. On line 74, we wrote:

From Results

We can reject the null hypothesis of a weakly positive effect at the 5% level ($p = 0.013$) for the June ENSO Outlook and nearly at the 10% level ($p = 0.11$) for the Winter Outlook.

As the referee requested, our text never claimed significance at the 10% level: we said “*nearly* at the 10% level” (emphasis added). So we believe we already meet the referee’s request (i).

While we appreciate where the referee is coming from, we sincerely disagree on request (ii). In recent years, the academic literature has spilled much ink discussing “p-hacking” (including in Nature journals¹). This term refers to the tendency for researchers to fiddle with statistical analyses until they barely clear p-value thresholds, driven by researchers’ belief that barely clearing a conventional p-value threshold substantially raises their publication chances over barely missing it. Such incentives distort academic research, both because the p-value thresholds are arbitrary to begin with (many statisticians never liked them) and because the incentives to clear them distort the results reported in the scientific literature. Analyses have found that, indeed, p-values published in the academic literature cluster just below the arbitrary cutoffs of 1%, 5%, and 10%, with holes just above the

¹See <https://www.nature.com/articles/d41586-019-00857-9>, <https://www.nature.com/articles/506150a>, and <https://www.nature.com/articles/s41562-016-0021>, among others.

cutoffs: clear evidence of “p-hacking”. In response, a number of prominent journals have banned the publication of p-values.

We reported one of those p-values that seems to be missing from the literature, as our p-value of 0.11 for the Winter Outlook missed the arbitrary 10% threshold by the thinnest of margins. We could have made any number of small tweaks to our analysis that would have shifted standard errors by the tiny amount required to generate a p-value just below 10% (e.g., using less flexible controls, changing which controls we include, changing how we cluster standard errors, or changing the number of days in the estimation window). However, we declined to do so because we do not believe it is right to engage in p-hacking. Instead, we reported the p-value from our benchmark specification and described it honestly as nearly significant at the 10% level. We strongly disagree that we should instead label this result as merely “not significant”: our result would be no more or less real if the p-value were 0.09 instead of 0.11. Treating a marginal change in p-value as changing a result from not-real to real would both violate our statistical training and reinforce researchers’ incentives to simply p-hack the next time around.

We hope that the referee appreciates our honesty in declining to p-hack. Further, we also hope the referee considers the body of evidence in our paper, which includes other analyses of the Winter Outlook that are easily significant at even a 1% threshold.

Referee 1: Minor Comment 4

Referee 1: Comment 4

4. The share of industry groups impacts is not informative as the information from outlook is not equally valuable for different industries. Typically, it is related to the sample, and consider those differences have been captured in the next figure, I would recommend to drop it.

Figure 2b We believe the referee is discussing Figure 2b. This is a central result of our paper. It shows strong evidence that the ENSO and Winter Outlooks affect industry groups throughout the economy, not just the handful one might think of as directly exposed to weather. The main text describes its construction:

From Results

It estimates average effects by industry group (defined by 4-digit NAICS, see Methods) and plots the share of industry groups with negative estimates. We would expect around 50% of industry groups to have a negative estimate by chance (as seen for the Hurricane Outlook), but average implied volatility falls in around 90% of industry groups upon NOAA’s release of either the June ENSO or Winter Outlook.

Here we are treating each industry group’s estimate as either positive or negative and tallying up the share of estimates that are negative. This panel intentionally abstracts from the intensity of effects on each industry group (which were incorporated into the other 5 panels of Figure 2) in order to test how many were affected (i.e., how many had a negative estimate). Of course, some negative estimates will arise by chance even if the outlook has no effect, and our statistical thresholds adjust for that possibility (here we easily clear even the 1% significance threshold in the two outlooks that have significant effects).

The referee objects to our binary classification of each industry group as affected or not. We have two possible interpretations of the referee’s objection and are not sure which is correct. First, the referee may object to the failure of the binary classification to account for the intensity of effects in each industry group. Fortunately, we already report such tests. If one wants to account for the intensity of effects across industry groups, then one should instead look at Figure 2a, which is essentially that weighted average. If one wants to look at the intensity of effects industry group by industry group, then one should look at Figures 2c–d. And if one wants to explore whether effects cut across many industry groups, then one should look at

Figure 2b. We have revised the text to explicitly tell readers that Figure 2a accounts for the intensity of effects:

From Results

(The results in Figure 2a implicitly account for the intensity of industry groups' effects.)

Second, the referee's reference to a sample leads us to think the referee may believe that we are sampling from firms. In fact, we use the entire population of firms with liquidly traded options. As we say in the Methods:

From Methods

Equity options data are from OptionMetrics. We use all firms available in IvyDB US (accessed through Wharton Research Data Services). This broad sample biases us towards finding no effect because many of these firms may not operate in the geographical regions of interest to the seasonal outlooks studied here.

We hope these revisions and explanations allay the referee's concern. And if we have misunderstood the referee's concern, then we invite a more detailed comment.

Referee 1: Minor Comment 5

Referee 1: Comment 5

5. It is confused to state 'effects of each month's ENSO Outlook'', we only know ENSO outlook is released in June. Once the information is released, all values will be in prices.

Figure 4a We believe the referee is commenting on Figure 4a, whose caption reads:

From Fig 4 Caption

Left: Estimated effects of each month's ENSO Outlook, as in Figure 2a. Black markers indicate that the estimate is significant at the 10% level.

The associated text reads:

From Results

Figure 4a estimates effects from each month's ENSO outlook. Consistent with the one-off jump in skill upon moving past the spring barrier, the June Outlook is the only release that shows significant effects at the 10% level.

We may be misunderstanding the referee's comment (in which case we apologize), but it appears as if the referee might be misinterpreting our analysis.

First, the ENSO outlook is released every month, whereas we believe the referee is claiming it is released only in June. We focus on the June outlook in our main analysis because that is widely known as the one that contains a big jump in skill due to moving past the "spring barrier". However, an ENSO outlook is released in every month of the year. In response to the other referee's request, we have moved background about seasonal outlooks from the Methods into the main text. We believe that this background will make it clear upfront that the ENSO outlook is released 12 times each year, not just in June. Our revised introduction now reads:

From Introduction

The ENSO seasonal outlook is released monthly. ENSO refers to sea surface temperature and wind anomalies in the eastern Pacific. The state of ENSO is often found to predict climate variables elsewhere in the world, including temperature and precipitation.²³ Each month's outlook reports the current state of ENSO and provides predictions out to 9 months. The June ENSO Outlook is known to be especially informative because it is the first to take advantage of the jump in skill after the "spring barrier".^{24–29} This jump in skill arises in part because contributing factors to ENSO are especially noisy in the spring.³⁰

Second, we do not know how to interpret the referee's comment that "once the information is released, all values will be in prices" since that is the whole point of our analysis! We rely precisely on the fact that new seasonal climate information is incorporated into prices (and that traders anticipate this effect before the outlook is released). Perhaps the referee thinks the ENSO Outlook is released in June with a prediction for each month and that our analysis then somehow tests for effects when these later months arrive? If this is the case, then our explanation just above should make it clear that this is not how our analysis is structured, and if this is not the case, we apologize for any misunderstanding but are honestly unsure how to interpret the referee's comment.

Finally, one may wonder what to expect for non-June outlooks given the spring barrier. Imagine that no new information about ENSO ever arrives after June, so that the July, August, etc outlooks would always be identical to the June outlook. As the referee says, "Once the information is released, all values will be in prices." We should then expect to find no effects of any post-June outlook. And this is precisely what we find! In reality, of course, there is some new information each month even if there is not as much new information as in June, but our tests suggest that markets do not expect this new information to be substantial in any given post-June month.

We hope our explanation clarifies our figure for the referee. If we have

misunderstood the referee’s concern, then we invite a more detailed comment.

Referee 1: Minor Comment 6

Referee 1: Comment 6

6. Did authors control for temporal variables such as weekdays as we find release days are not always the same?

Controls The referee asks whether we control for temporal variables. Yes, we do. In fact, our preferred specification includes many more temporal controls than does the run-of-the-mill event study. In the Methods, we write:

From Methods

X_{it} is a vector of controls, with Γ_{iy} a coefficient vector and $\mathbb{1}_{t \in y}$ an indicator for whether trading day t is in year y . This specification of the regression equation allows the effects of the controls to vary by firm and by year. In the authors’ preferred specification, the controls include a constant, the option’s time to expiration and its square, the log-change in the London Inter-Bank Offered Rate (LIBOR, which was a benchmark interest rate tied to prominent banks’ willingness to lend to each other), the log-change in the 10-year Treasury rate, and dummy variables for a 3-day event window centered on any earnings announcements for firm i (we drop any firm-outlook-year triplet with an earnings report in the 3-day event window).

We are using a strong set of controls, well beyond what is usually required of event study analyses. And we implement them in an especially flexible fashion, as we allow their effects to vary with each firm and each year. Remarkably (and reassuringly!), Figure S-4 in SI E shows that our results are virtually unchanged even if we drop all of these controls.

In order to clarify in the main text that our framework includes temporal controls, we have added the following text to the Results:

From Results

Our primary results are event study estimates. Our event study framework removes news related to factors such as interest rates and earnings reports and then tests whether the remaining news on the days a seasonal outlook is released is unusual relative to other days in the sample (see Methods).

The referee also asks about day-of-the-week controls. Such controls would be highly unusual in this context. First, there has been a paper or two discussing reasons why one might expect a “Monday” effect and there have been a few tests of such an effect, but this is not a widely accepted effect and we know of no theoretical or empirical work suggesting effects from other weekdays. (Note: Table S-2 in SI B shows that our outlooks of interest were not released on a Monday.) Second, a weekday control would reduce our effective sample size dramatically: we use a 30-day estimation window, so each day-of-week control would be identified off only 6 observations. Third, Table S-2 in SI B shows that each outlook tended to be released on a consistent day of the week in 2010–19.

If the referee would like to see some other temporal control, then we are happy to implement it if feasible. However, we do believe that our existing temporal controls are rather strong and, given the robustness of our results to dropping all of these controls, we would be surprised if an additional control would change our results.

Referee 2

Thank you for your helpful comments and guidance. We describe how we have addressed all of your concerns below.

Your comments are below, `boxed in with a gray background and different font`. My responses are in plain text. Any included text from the new manuscript is `boxed in with a white background and the` `same font as this text` with the section number noted at the top of the quoted text.

Referee 2: Comment 1

Referee 2: Comment 1

The aim of this article is to study the value financial markets attach to longer-term weather forecasts (mainly the ENSO outlook by NOAA).

The analysis seems competently conducted (with the caveat I have not tried to replicate the results, nor I have gone through the technicalities in detail). It reads as a solid piece of work from an empirical and financial perspective. In principle, I would be in favour of seeing this published in Nature Communications.

However, I would like to suggest two main things to the authors.

First, I think the writing style could be improved by making things clearer, especially to the reader unexperienced with finance. Certain important bits of info are relegated to the Methods section: the most prominent example is the very first paragraph of Methods, where (finally!) authors offer readers the clear synthetic info of what they're talking about. This is not 'methods', but rather essential info the reader needs to reads to understand from the very start so to appreciate what comes next. I was rather confused on some details explained only later in the text (e.g. which outlooks are you analyzing; or what do you mean by 'skill'). If I'm not mistaken, even ENSO is never really explicitly defined (if one doesn't know what it is, why should they care?). I can see the authors already put an effort in explaining options and other concepts in clear terms, but I feel they are not there just yet.

Organization We appreciate this comment. We have worked to tighten the writing throughout, including by rewriting the conclusions. The referee has a few specific requests.

First, background material about seasonal outlooks was previously in our Methods and SI A. We have now brought more of this background into the very beginning of the manuscript, with text that now reads:

From Introduction

We test whether options markets in 2010–2019 priced uncertainty about the news contained in upcoming seasonal climate outlooks for winter weather, the El Niño Southern Oscillation (ENSO), and hurricanes. The U.S. National Oceanic and Atmospheric Administration (NOAA) releases each seasonal outlook on a regular, announced schedule with strict rules to guard against information leaking in advance (see Supplementary Information B). Each product provides multi-month predictions (and hence is a “seasonal” outlook). “Skill” refers to the accuracy of an outlook’s forecasts over many years. The Atlantic Hurricane Seasonal Outlook is released in May and includes outlooks for both the Atlantic and Eastern Pacific basins. The U.S. Winter Outlook is typically released on the third Thursday of October. It reports the probability that each part of the country will experience abnormal seasonal temperatures or precipitation over the coming December through February. The ENSO seasonal outlook is released monthly. ENSO refers to sea surface temperature and wind anomalies in the eastern Pacific. The state of ENSO is often found to predict climate variables elsewhere in the world, including temperature and precipitation.²³ Each month’s outlook reports the current state of ENSO and provides predictions out to 9 months. The June ENSO Outlook is known to be especially informative because it is the first to take advantage of the jump in skill after the “spring barrier”.^{24–29} This jump in skill arises in part because contributing factors to ENSO are especially noisy in the spring.³⁰

Second, the referee asked us to clarify which outlooks are being used, what we mean by “skill”, and what ENSO is. We believe the previously mentioned introduction paragraph addresses all of these concerns. In particular, we define “skill”:

From Introduction

“Skill” refers to the accuracy of an outlook’s forecasts over many years.

And we define ENSO:

From Introduction

ENSO refers to sea surface temperature and wind anomalies in the eastern Pacific. ENSO is often found to predict climate variables elsewhere in the world, including temperature and precipitation.²³

Finally, we have added more intuitive descriptions of our statistical framework. We have added a short description to the main text:

From Results

Our primary results are event study estimates. Our event study framework removes news related to factors such as interest rates and earnings reports and then tests whether the remaining news on the days a seasonal outlook is released is unusual relative to other days in the sample (see Methods).

And we have reorganized our Methods to include a first section labeled “Experimental Design”. It aims to give our readers a high-level sense of our statistical framework before delving into the details of data, regression equations, and hypothesis testing:

From Methods

We use event studies to detect whether option prices incorporate information about seasonal outlooks. Event studies isolate movements in financial variables due to the news released on particular days. They test whether the news released on days of interest is unusual relative to the news released on other days, after removing types of news that are explained by the controls. In our case, the event study removes news related to factors such as interest rates and earnings reports and then tests whether the remaining news on the days a seasonal outlook is released is unusual relative to other days in the sample.

We seek to test the hypothesis that releasing an outlook reduces uncertainty. However, we cannot test this hypothesis by examining changes in option prices: the specific contents of an outlook release also affect option prices by changing the price of the underlying stock, and as a result of this effect and of changes in price due to changing time to expiration, the average change in option prices should be zero (under the risk-neutral measure, by familiar no-arbitrage arguments). We therefore test our hypothesis by testing whether the release of an outlook reduces options' implied volatility. Analyzing implied volatility is, in effect, a nonlinear way of controlling for the effects of changes in the stock price and time to expiration. Our specific methodological approach follows previous literature in running an event study in implied volatility.^{56–63}

As detailed in Supplementary Information D, the effect of seasonal outlooks will be identified as long as other news that affects uncertainty about future stock prices is not systematically paired with the release of seasonal outlooks over the course of the decade. Of course, there will always be other news on any given day. We just need the news paired with outlook releases over the years to be as-good-as-random, net of our many controls. Supplementary Information D shows that obvious candidates for such news are not likely to be a problem, and robustness checks in Supplementary Information E show that our results are not sensitive to statistical specification.

We hope that this additional effort clarifies our approach for readers with diverse disciplinary backgrounds.

Referee 2: Comment 2

Referee 2: Comment 2

Second, and probably even more important, I think the authors should present better why this analysis is relevant and interesting. Financial markets put some value in the uncertainty on ENSO and its resolution. So what? Why should I be interested in this? Is the value particularly high or low? Does this mean something special? While I appreciate the competence of the analysis, I am slightly struggling in understanding what to do with it. Haven't weather outlooks always been relevant for markets? I think the authors are trying it to connect the topic to climate risks in the discussion section, but I didn't find it compelling. I also struggle to perceive this study to be about climate change, as the authors seem to suggest. Is the study about the change in ENSO frequency or strength? If not, isn't the paper only about climate, per se? ENSO and hard winters have been happening also before anthropogenic change.

Relevance of Analysis We hear the referee asking for a clearer upshot of the study. We have made several changes in order to address this concern:

1. The leading weather agencies are focusing on improvements to seasonal outlooks, but we currently know next to nothing about whether they are currently valuable to society. In the first paragraph, we have added concrete details:

From Introduction

The U.S. Weather and Research Forecasting Innovations Act of 2017 elevated seasonal forecasting innovations to one of the National Weather Service’s five focus areas,² and the European Centre for Medium-Range Weather Forecasts’ 2021–2030 strategy highlights producing skillful seasonal outlooks as one of the four outcomes that indicate progress in meeting user needs.³

We then immediately give reasons why we may doubt that such improvements should be a priority:

From Introduction

Yet water resource managers have not relied on seasonal forecasts and would not prioritize their further improvement,^{4,5} and there are theoretical reasons to believe that the existence of skillful short-run forecasts undercuts the value of longer-run seasonal forecasts.⁶ Many have lamented that policy priorities must be developed without knowing society’s current value for forecasts or how that value would increase if forecasts became more skillful.^{7–9}

And we shortly thereafter say why we need to know how traders value seasonal outlooks:

From Introduction

Learning whether traders do indeed value longer-run forecasts and which economic sectors they see seasonal forecasts affecting should inform how governments allocate funds towards producing and improving seasonal forecasts.

2. We have added a new paragraph to the top of the Discussion that directly frames our results in terms of policy priorities:

From Discussion

Meteorological agencies currently prioritize improvements in seasonal forecasting,^{2,3} but it has been an open question whether seasonal forecasts are currently useful to the private sector. We provide the first real world measure of the value of publicly funded seasonal outlooks. Our novel research design shows that financial options markets do price uncertainty about the contents of upcoming ENSO and winter seasonal outlooks: traders must believe that these seasonal outlooks may contain information relevant to firms' future performance. Importantly, our measures are lower bounds on outlooks' total value to society, as we do not measure nonmarket benefits, we ignore firms lacking liquidly traded options, and we do not measure the losses avoided by adaptation that uses outlooks' contents. Moreover, we measure only the incremental effect of seasonal outlooks against background information: some of the information in a given outlook will already be available through non-NOAA forecasting efforts, agents may extrapolate recent months' ENSO forecasts to generate current forecasts of ENSO, hurricane activity, or winter weather, and some of the contents of the Winter Outlook are used in the Energy Information Administration's Winter Fuels Outlook that is often released 1–2 weeks earlier. The total value of advance information about seasonal climate should include the value of this background information.

And the next paragraph then gets into some more detailed conclusions about the Hurricane Outlook:

From Discussion

The fact that we detect sizable effects despite these limitations suggests that advance information about winter and ENSO seasonal climate is rather valuable. And the fact that we do not detect such effects from the Hurricane Outlook suggests that this outlook may need to become more skillful and/or more specific if it is to provide value over and above shorter-run forecasts of particular storms. In fact, there are scientific efforts towards building future seasonal outlooks of U.S. land-falling hurricanes instead of the current operational basin-wide numbers (which may hit the U.S. or not).³⁸ As hurricane outlooks advance in skill or are upgraded to include regional landfall, their impact on markets may change as it becomes easier to translate physical climate information (such as number of storms) to financial impact.

3. Beyond the value of outlooks, economists have by and large studied the effects of localized weather events but not of larger-scale weather patterns, which stretch over time and space. Our results imply that these patterns are important in their own right. We have added a paragraph on this topic to the Discussion:

From Discussion

Beyond the utility of seasonal outlooks, it has also been an open question whether seasonal climate patterns (as opposed to particular weather events in particular places) matter to the private sector. Such patterns are difficult to analyze because they affect weather in a variety of ways and over multimonh timescales. The release of a seasonal outlook collapses all these dimensions and timescales of realized weather into a discrete change in information on a particular day. Our results imply that seasonal climate patterns do matter—and they matter for firms throughout the economy. Future work should expand the economic analysis of weather beyond localized events to consider the import of large-scale atmospheric patterns.

4. Climate change is an implication, not our main point. Accordingly, we have removed the climate change reference from the abstract and made it clearer in the Discussion that our work has two suggestive implications for climate change, even though it is not about climate change. And in the discussion of risks, we have both tightened the material and added the following implication:

From Discussion

Moreover, traders expect seasonal climate to affect firms throughout the economy, not just firms in sectors that are obviously exposed to weather. This broad exposure to seasonal climate risk suggests the potential for broad exposure to long-run climate risks.

We thank the referee for spurring us to address the importance of our study more forthrightly. We hope these revisions satisfy the referee and welcome further suggestions.

REVIEWER COMMENTS

Reviewer #3 (Remarks to the Author):

“Financial Markets Value Skillful Forecasts of Seasonal Climate”

The authors use financial options markets to value the information provided by seasonal forecasts. The forecasts are the June ENSO El Nino forecast, the Winter Weather forecast, and the seasonal Atlantic Hurricane Outlook, all by NOAA, as well as the Farmers Almanac winter outlook and the Colorado State Hurricane Outlook.

A call option is the right, but not the obligation, to buy a stock at a fixed price at a future date. As the stock price rises, the option becomes more valuable as the fixed price is less than the increasing market price. But as the stock price falls below the fixed price, it becomes cheaper to buy the stock at the market price and so the option has no value. For this reason, the value of the option ex ante is increasing in the variability of the stock price, as the option owner benefits from upside risk but is protected from downside risk. The price of the option can be expressed as a function of the variance via the famous Black-Scholes formula. By inverting this formula, one can derive the implied volatility/variance as a function of the observed option price.

Thus the author strategy is to look at the change in implied volatility following the release of a forecast. If the implied volatility falls, then the forecast has reduced the volatility of the stock price, and thus provided useful information.

The authors find that (1) implied volatility drops following the release of only the El Nino and NOAA winter forecasts, (2) the drop in implied volatility has increased over time, (3) market participants are willing to pay about \$12 million to insure the risk associated with an upcoming El Nino forecast, and (4) the drop in volatility varies by industry, for example construction and recreation see large drops, but unexpectedly agriculture and insurance see insignificant effects.

General Comments

The paper is very well done and introduces a plausible and interesting new methodology (using options) to a long standing and important problem (the value of forecasts). That being said,

I have a few comments.

The only major concern I have is the fixed timing of the forecast releases. Consider the ENSO release in June. If the release date corresponded to the release of, say, a macroeconomic statistic such as inflation or unemployment, or even corresponded to the release of important firm level data such as “earnings season,” then the drop in volatility might be in fact caused by the release of economic data. I checked briefly and for example the CPI and other inflation data is released the week after ENSO and jobless claims is released on the same day (in 2023). June is also earnings season. It would be helpful for the reader if the authors took a careful look at major releases to make sure they are not the cause of the observed drop in implied volatility.

Specific Comments

Abstract, “little is known” seems a little strong. Also, page 13, “the first real world measure...”. The use of options is innovative and interesting, but there are lots of papers on the value of seasonal forecasting. Meza, et. al., *Journal of Applied Meteorology and Climatology* (2008) reviews some 33 studies and there are many others.

Abstract, line 19, a third possibility is that initial exposure was extremely high and ex ante adaptation reduced this exposure to lower, but still high level, at which point FURTHER adaptation is costly/limited.

Page 2, line 33 another example of valuing forecasting using real markets is through prediction markets. See for example, Kelly, et. al. *JEBO* (2012). This method has the advantage of removing non-weather related volatility.

Page 2, line 34, one can view a climate model as a type of long term forecast. Schlenker and Taylor, *Journal of Financial Economics* estimate the effect of model-climate forecasts on weather futures contracts. Since weather futures contracts also price risk, it would seem useful to compare vs the options approach used here.

End of page 4, Roll *AER* (1984) is one of the seminal papers looking at the effect of forecasts on derivative prices. Yes, they consider only short term forecasts in a single market. Still, it would be interesting to hear the authors’ view about this approach versus their use of options markets.

As the authors note on page 5, line 84, the release of new information should decrease uncertainty on average. But consider CO forecast of the number of hurricanes. We might think the volatility might remain high or even increase if the number of forecasted hurricanes is high, as there is now more

uncertainty about the number of hurricanes making landfall. Similarly, if less than average hurricanes were forecasted, we might expect a large decline in volatility. Could we improve the results by looking at the decline in volatility as a function of the number of forecasted hurricanes instead of just the release date? Perhaps the CO forecast is not informative on average, but is informative if the forecast is for few hurricanes, for example.

Page 7-8, it is a little disconcerting that agriculture is the industry with the smallest/most insignificant point estimates, and that mining and insurance have small effects. Probably the industry effects are not significantly different from each other, although this also seems difficult to explain. As the authors state, general equilibrium effects is a possible cause. But another possible cause would be the release of some macro data at a similar time that affects many industries.

Figure 3a-b, I'm not sure "total market cap exposed" is the right terminology. Perhaps "total market cap of companies with some exposure"? Obviously there is no way an El Nino event could ever cause \$16 trillion of impacts. The regression results give the difference in risk before versus after the forecast release. But it is not clear how one could recover the baseline El Nino risk from the difference. In any event, it would seem more important to focus on the premium and risk reduction numbers rather than the market cap.

Page 23, line 398, it would seem that the binomial tree model does not account for fat-tailed risk in securities prices. It might be helpful to point this out.

Reviewer #4 (Remarks to the Author):

This paper is potentially a very important demonstration of the aggregate economic impact and value of seasonal forecasts, which previously has not been quantified except in comparatively limited case studies. In my opinion it will be a valuable and doubtless highly cited addition to this literature, although certain details need further addressing and/or explaining as outlined below.

I should emphasize before continuing that my perspective is that of a climate scientist with expertise in seasonal predictability and prediction, but not the financial data and analysis methods that are at the core of this study. Therefore it is essential that the paper also be reviewed (or have been reviewed) by someone familiar with the methods applied to assess the impact of anticipated information on option volatility, as exemplified in references 56-63.

That said, the simple illustrative example in Fig. 1 is nicely crafted and helpful to the non-specialist for understanding how seasonal outlooks potentially can influence option prices by reducing uncertainty.

Specific comments:

1) The presented results appear for the most part to be solid statistically, with some possible exceptions that are discussed below. However, as for many statistical analyses, various choices are made that potentially could channel the outcomes toward spurious statistical significance if the authors allowed themselves to be so influenced. On one hand, their arguments for the singular importance of the June ENSO outlook are plausible. On the other hand, it is not obvious why, for example, the authors focus on “an October–December target as that window is especially informative about boreal winter conditions” (lines 495-496), considering that (a) such a window mostly precedes boreal winter, (b) the sea surface temperature manifestations of ENSO tend to be strongest in November to January, and (c) ENSO’s impacts on North American climate tend to be most pronounced after December, in the mid-to-late winter and early spring months. Could the authors provide a more specific and robust argument for this choice, and comment on sensitivity to considering alternative targets such as November-January or December-February?

2) Regarding the relative ordering of sectoral impacts on the June ENSO vs Winter Outlooks shown in Fig. 2c-d and discussed near line 125, would it be worth noting that ENSO impacts are global, whereas the Winter Outlook pertains to the United States, and that different sectors likely have different global vs national seasonal climate exposures?

3) The effects in Fig. 4b-c associated with the CSU Hurricane Outlook are significant at $p < 0.05$, but opposite in sign to those associated with the June ENSO Outlook. Do the authors have any hypotheses for why this is, other than it being a statistical fluke?

4) Can the authors explain why the effects in Fig. 4b are far more statistically significant than those in Figs. 4a and c, particularly as this implies effects of the May and July outlooks that are hard to explain in terms of their hypotheses? Could these uncertainties perhaps have been systematically underestimated? (If so then this bears on Fig. 3b as well.)

5) Regarding Fig. 4a, it is expected that 1 or 12 cases would be significant with $p \leq 0.1$ purely by chance. Therefore the (marginal) significance of this result is that it occurs for the June Outlook, in accord with the authors’ hypothesis.

6) The ENSO outlook skill data that is described as being provided in reference 37 is presented there in a graphical form that is not readily translated into precise numerical values. Therefore the authors must have obtained the numerical data from the authors of that paper, or else have reproduced their calculations. Could this be clarified? (Possibly this is addressed in the Supplementary Information, which I did not have access to.)

7) Reference 31 needs to be updated.

8) Some final comments: From the perspective of seasonal forecast providers, this study answers some questions but raises others, particularly in relation to the “scientific literacy” of the markets. Given the apparent economic advantages involved, it may not be surprising that the markets apparently are aware of the relatively large skill jump from the May to June ENSO outlook, for example, and can differentiate between more and less skillful and trustworthy outlooks which may have similar visibility. An interesting question would be how such knowledge is transmitted from the science to the financial communities. For example, is there a small number of climate experts in the financial world whose knowledge diffuses throughout the community to become widely known rules of thumb? Another aspect is that the official outlooks seldom hold surprises for knowledgeable members of the scientific and perhaps financial communities, who through monitoring and experience will be able to anticipate the broad character of the outlooks weeks to months before they are released. (This is broadly addressed in lines 208-214.) Finally, although the effects described in this paper align with widely accepted science, it is worth being alert to exceptions. One possible exception is that reference 20 found markets to have apparently priced in the Francis and Vavrus (2015) suggestion that anthropogenically-driven changes in the jet stream are causing an increased incidence of cold winters in the eastern US, given that this view has remained controversial.

Referee 3

Thank you for your helpful comments and guidance. We do our best to answer all of your questions below.

Your comments are below, `boxed in with a gray background and different font`.

My responses are in plain text. Any included text from the new manuscript is `boxed in with a white background and the`
`same font as this text` with the section number noted at the top of the quoted text.

Referee 3: Overview

Referee 3: Overview

The authors use financial options markets to value the information provided by seasonal forecasts. The forecasts are the June ENSO El Nino forecast, the Winter Weather forecast, and the seasonal Atlantic Hurricane Outlook, all by NOAA, as well as the Farmers Almanac winter outlook and the Colorado State Hurricane Outlook.

A call option is the right, but not the obligation, to buy a stock at a fixed price at a future date. As the stock price rises, the option becomes more valuable as the fixed price is less than the increasing market price. But as the stock price falls below the fixed price, it becomes cheaper to buy the stock at the market price and so the option has no value. For this reason, the value of the option ex ante is increasing in the variability of the stock price, as the option owner benefits from upside risk but is protected from downside risk. The price of the option can be expressed as a function of the variance via the famous Black-Scholes formula. By inverting this formula, one can derive the implied volatility/variance as a function of the observed option price.

Thus the author strategy is to look at the change in implied volatility following the release of a forecast. If the implied volatility falls, then the forecast has reduced the volatility of the stock price, and thus provided useful information.

The authors find that (1) implied volatility drops following the release of only the El Nino and NOAA winter forecasts, (2) the drop in implied volatility has increased over time, (3) market participants are willing to pay about \$12 million to insure the risk associated with an upcoming El Nino forecast, and (4) the drop in volatility varies by industry, for example construction and recreation see large drops, but unexpectedly agriculture and insurance see insignificant effects.

Referee 3: Overview (Continued)

General Comments

The paper is very well done and introduces a plausible and interesting new methodology (using options) to a long standing and important problem (the value of forecasts). That being said, I have a few comments.

We appreciate the thoughtful summary and kind evaluation.

Referee 3: Major Comment 1

Referee 3: Major Comment 1

The only major concern I have is the fixed timing of the forecast releases. Consider the ENSO release in June. If the release date corresponded to the release of, say, a macroeconomic statistic such as inflation or unemployment, or even corresponded to the release of important firm level data such as ‘earnings season,’ then the drop in volatility might be in fact caused by the release of economic data. I checked briefly and for example the CPI and other inflation data is released the week after ENSO and jobless claims is released on the same day (in 2023). June is also earnings season. It would be helpful for the reader if the authors took a careful look at major releases to make sure they are not the cause of the observed drop in implied volatility.

The referee is concerned that important economic news may have systematically been released on the same days as the seasonal outlooks. We agree that this is the major threat to identification. In the prior submission, we addressed this concern in Section D of the SI, where we collected the dates of several major news releases (Federal Reserve minutes, GDP estimates, and employment reports) and showed that they did not coincide with the

outlooks' release dates.

In response to the referee's comment, we have collected the dates of several more news releases: the Consumer Price Index, the Producer Price Index, and a few additional employment reports. The reorganized Table S-3 includes these dates as well as the ones from our prior submission. We copy that table here as Table R1. We have expanded the associated text in the SI as follows:

From SI D

Second, one might be concerned about several of the most prominent economic announcements made by the Federal Reserve, the Bureau of Labor Statistics, and others. Table S-3 reports the release of major economic news within three days of a seasonal outlook release. It considers seven different types of news releases, which together capture the major interest rate, GDP, employment, and inflation announcements. Entries with a 0 (in bold) indicate that an announcement came out the same day as the NOAA outlook. We see that this coincidence never happened for the June ENSO Outlook, happened five times for the Winter Outlook in 2000–2019 and three times for the Winter Outlook in our 2010–2019 period of interest (with two times being inflation reports and one time an unemployment report), and happened one time for the Hurricane Outlook (an unemployment report). Our June ENSO estimates will not be affected by any of these reports, and our other estimates may not be either.

The referee also asked about jobless claims data. These dates are not in Table S-3 because this is a weekly release that overlaps with nearly every NOAA outlook release we study. However, the fact that this release overlaps with nearly every one of the various NOAA outlooks is itself evidence that it is unlikely to drive our results, as the we describe in the following text that we added to SI D:

Table R1: Economic News Released within One Week of NOAA Outlooks, 2000–2019

Year	June ENSO	Winter	Hurricanes
2000	N/A	+1(P)	-3(E),+2(P)
2001	+3(P)	+2(C)	-3(U1)
2002	+1(E)	+1(C)	-3(U1)
2003	+1(P)	0(C)	-3(C)
2004		-2(C),+1(U1)	-3(C)
2005		+1(U1)	+1(P),+2(C)
2006		-2(P),-1(C),+1(U1)	-2(U1)
2007		-2(E),+3(P)	
2008	+1(E)	-2(P),-1(C),+1(U1)	-2(P)
2009	-1(U2),+1(E)	0(C)	+1(U1)
2010	-1(U2),+1(E)	+1(U1)	
2011		-2(P),-1(C),+1(U1)	+1(U1)
2012		-2(C),+1(U1)	
2013	+1(E)	-1(C), 0(P) ,+1(U1)	
2014	+1(E)	-1(P)	
2015	+1(P)	-1(P), 0(C)	0(U1)
2016		-2(C),+1(U1)	
2017		+1(U1)	
2018	-2(C),-1(F,P),+1(U1)	+1(U1)	
2019	-2(P),-1(C)	0(U1)	

Numbers in the table signify how many days before (-) or after (+) the NOAA outlook was the economic outlook released. Bolded values were released on the same day as the NOAA outlook. Cells are empty if none of the economic outlooks was released within 3 days on either side of the NOAA outlook.

C: Consumer Price Index, Real Earnings

E: Employment Situation Summary

F: Federal Open Market Committee Press Release

G: GDP Growth Estimate

P: Producer Price Index

U1: Regional and State Employment and Unemployment

U2: Metropolitan Area Employment and Unemployment

From SI D

Third, one might also be concerned by releases of the Unemployment Insurance Weekly Claims Report, which includes initial jobless claims. In our study period, these releases occur on Thursdays. Table S-2 shows that all three NOAA outlooks were nearly always released on a Thursday. If initial jobless claim news were driving our estimated effects then we would expect to detect similar effects for all of the NOAA outlooks. However, we do not in fact detect a significant effect for the NOAA Hurricane Outlook. In addition, all months' ENSO outlooks were released on Thursdays, not just the June outlook (see Section B). But Figure 4a in the main text showed that we detect a significant effect only for the June ENSO Outlook. This evidence again makes it unlikely that anticipation of weekly initial jobless claims data drives our results.

The referee also wondered about earnings reports. The possibility of such a correspondence was indeed a concern for us. As a result, we have always included dummies for the days around an earnings report release. These dummy variables essentially remove firm-day pairs with (or even near) earnings releases from our estimation sample. Moreover, we drop any firm-year pairs with an earnings report within one day of an outlook release. In the main text, we wrote:

From Results

Our event study framework removes news related to factors such as interest rates and earnings reports and then tests whether the remaining news on the days a seasonal outlook is released is unusual relative to other days in the sample (see Methods).

In the Methods, we had described the relevant controls:

From Methods

dummy variables for a 3-day event window centered on any earnings announcements for firm i (we drop any firm-outlook-year triplet with an earnings report in the 3-day event window).

We have added material on this point to our SI Section D, which is focused on identification:

From SI D

First, investors may anticipate earnings reports that come out around the same time as NOAA outlooks. We take care of this concern in two ways (see Methods). First, when estimating effects for any given outlook, we drop firm-year pairs that have an earnings report within a day of the outlook release. Second, we dummy out the day a company's earnings are reported as well as the day before and after it. So any company-year pairs with an earnings report within a day of a NOAA outlook do not contribute to the estimated effect, and any company-day pairs within a day of an earnings report also do not contribute to the estimated effect.

We hope the referee is satisfied that we did due diligence in checking for correspondence between NOAA releases and a range of economic releases. We also hope the referee is satisfied that weekly jobless claims data cannot explain the differential effects we detect across outlooks and that we had adequately addressed the possibility of co-occurring earnings releases. We are happy to collect the dates of other information releases if the referee is concerned about specific additional ones.

Referee 3: Specific Comment 1

Referee 3: Specific Comment 1

Abstract, ‘‘little is known’’ seems a little strong. Also, page 13, ‘‘the first real world measure...’’. The use of options is innovative and interesting, but there are lots of papers on the value of seasonal forecasting. Meza, et. al., *Journal of Applied Meteorology and Climatology* (2008) reviews some 33 studies and there are many others.

We appreciate these points. We were (unsuccessfully) attempting to distinguish revealed preference from simulation methods, but our language was indeed too strong. We have edited that language in the abstract to ‘‘but they do so without much information about these forecasts’ value in practice.’’ This language better matches the lament we saw in the literature and described as:

From Introduction

Many have lamented that policy priorities must be developed without knowing society’s current value for forecasts or how that value would increase if forecasts became more skillful.^{7–9}

We have edited the language on the former pg 13 (‘‘We provide the first real world measure of the value of publicly funded seasonal outlooks.’’) to:

From Introduction

We measure the value of publicly funded seasonal outlooks to financial market participants.

And we have added the Meza et al. review to the citations for prior work using decision models.

Referee 3: Specific Comment 2

Referee 3: Specific Comment 2

Abstract, line 19, a third possibility is that initial exposure was extremely high and ex ante adaptation reduced this exposure to lower, but still high level, at which point FURTHER adaptation is costly/limited.

Thank you for raising this point. Your description was the kind of outcome we had in mind, but it got lost in the concise phrasing. In addition, we also want to permit the possibility described in the Discussion, wherein firms do eliminate exposure but the adaptation itself is costly. The old sentence read: “Because we estimate substantial exposure to the forecasted portion of seasonal climate, ex-ante adaptation to seasonal climate must either be costly or be limited in scope.” We have revised it to read:

From Abstract

Firms must not be able to undertake ex-ante adaptation that would eliminate their exposure to the forecasted portion of seasonal climate without imposing substantial costs of its own.

We elaborate further in the Discussion. We appreciate the suggestion and hope the referee finds the new language to be more precise.

Referee 3: Specific Comment 3

Referee 3: Specific Comment 3

Page 2, line 33 another example of valuing forecasting using real markets is through prediction markets. See for example, Kelly, et. al. JEBO (2012). This method has the advantage of removing non-weather related volatility.

Thank you for pointing us to this paper that we had not found earlier. We have edited the following sentence to specifically include prediction markets and cite this paper:

From Introduction

Other work showed that financial or prediction market participants attend to more conventional weather forecasts that have horizons of days^{19–23} ...

In terms of the relation to our approach, prediction markets could act like options markets if there were predictions for each interval in a span, so that analysts could reconstruct probability density functions from prediction markets. In that case, one could test for changes in implied variance following the methods used here.

Referee 3: Specific Comment 4

Referee 3: Specific Comment 4

Page 2, line 34, one can view a climate model as a type of long term forecast. Schlenker and Taylor, Journal of Financial Economics estimate the effect of model-climate forecasts on weather futures contracts. Since weather futures contracts also price risk, it would seem useful to compare vs the options approach used here.

This is a good point. We previously mentioned this paper's work on short-run weather forecasts, and we now cite this paper's work with climate model output as well. We have modified the text to read:

From Introduction

Other work showed that financial or prediction market participants attend to more conventional weather forecasts that have horizons of days^{19–23} and even to climate model forecasts of multi-year trends.²³

In terms of the relation to our approach, the futures contracts used in that paper are the analogue of using stock prices, an approach we describe in the introduction and in Figure 1. One could test for jumps in the futures contracts used by that paper when a particular seasonal outlook is released. Such tests would be interesting for teasing out the perceived implications of that seasonal outlook for weather in particular locations, but if we averaged over many releases of a seasonal outlook, we would expect to find no change in the futures price on average. As we write in the introduction,

From Introduction

Testing for these stock price reactions would reveal whether a particular year's forecast was both surprising relative to expectations and relevant for profits. However, stock price reactions do not tell us how markets value the regular production of outlooks because different years' outlooks should not affect stock prices on average.

We have now added a subsequent sentence,

From Introduction

Some prior work studies effects of forecasts in futures markets;^{19–21,23} for our purposes, futures prices act like stock prices, as they average over possible future outcomes.

If we had options on those same weather events, then we could adopt the approach of the present paper to detect how weather markets value the regular production of seasonal outlooks.

Referee 3: Specific Comment 5

Referee 3: Specific Comment 5

End of page 4, Roll AER (1984) is one of the seminal papers looking at the effect of forecasts on derivative prices. Yes, they consider only short term forecasts in a single market. Still, it would be interesting to hear the authors' view about this approach versus their use of options markets.

We should have included a citation to Roll in the following sentence and have now corrected that omission:

From Introduction

Other work showed that financial or prediction market participants attend to more conventional weather forecasts that have horizons of days^{19–23}

Regarding its methods' relation to our paper, please see our prior reply for why futures markets are distinct from our approach and for the modifications we have made to highlight this difference. We cite Roll in the text we added as part of the prior reply:

From Introduction

Some prior work studies effects of forecasts in futures markets,^{19–21,23} for our purposes, futures prices act like stock prices, as they average over possible future outcomes.

Referee 3: Specific Comment 6

Referee 3: Specific Comment 6

As the authors note on page 5, line 84, the release of new information should decrease uncertainty on average. But consider CO forecast of the number of hurricanes. We might think the volatility might remain high or even increase if the number of forecasted hurricanes is high, as there is now more uncertainty about the number of hurricanes making landfall. Similarly, if less than average hurricanes were forecasted, we might expect a large decline in volatility. Could we improve the results by looking at the decline in volatility as a function of the number of forecasted hurricanes instead of just the release date? Perhaps the CO forecast is not informative on average, but is informative if the forecast is for few hurricanes, for example.

We believe the referee is suggesting that the zero lower bound on hurricanes mechanically makes predictions for more storms have greater variance. In that case, perhaps we would detect a decline in variance for the Colorado State University hurricane outlooks if we focused on the release of their outlooks that, ex post, happened to forecast a low-storm (or otherwise low-variance) seasonal climate. The referee observes that such a test might show that markets do pay attention to the contents of some particular CSU outlooks, even if they don't expect much value from these outlooks on average.

We do not disagree with this observation. In many ways, the referee's proposed test ends up being similar to tests of equity or futures market responses to the release of outlooks. Such a test can tell us whether markets pay attention to the news contained in a particular outlook. In the case of equity or futures markets, we learn whether the particular outlook's news

changed expected weather. As we understand it, the referee's proposed test would tell us whether the particular outlook's news changed agents' perceived variance of weather.

All such tests are certainly interesting, but they do not get at the question of interest here. We study not whether markets respond to particular forecasts but whether they value the production of forecasts, before knowing precisely what the forecast will say but while knowing that it will say something. For instance, we want to include the fact that traders do not know in advance whether the CSU Hurricane Outlook will forecast few or many storms, rather than conditioning on ex post information about what the outlook happened to forecast in practice.

In our discussion of the CSU outlook, we have added text that clarifies that our results do not imply that the outlook contains no information:

From Results

Our results do not imply that markets never respond to information contained in particular releases of the two non-NOAA outlooks, but they do imply that markets do not expect these outlooks to contain relevant information on average.

We hope that this change addresses what we assume is the referee's underlying concern: that a reader may conclude that these outlooks are never informative, whereas all we show is that markets do not expect information from them.

Referee 3: Specific Comment 7

Referee 3: Specific Comment 7

Page 7-8, it is a little disconcerting that agriculture is the industry with the smallest/most insignificant point estimates, and that mining and insurance have small effects. Probably the industry effects are not significantly different from each other, although this also seems difficult to explain. As the authors state, general equilibrium effects is a possible cause. But another possible cause would be the release of some macro data at a similar time that affects many industries.

We understand that the referee finds some of the least-affected industries to be counterintuitive and is concerned that these effects may indicate that we are actually picking up the release of macro news that happens to covary with seasonal outlook releases. We hope that our above response to the referee's major comment assuages such concerns. Regarding agriculture, it is important to remember that we find significant effects for the Winter Outlook and for the June ENSO Outlook. Both of these outlooks primarily forecast winter weather. So it may not be surprising that agriculture would not be especially exposed to these outlooks. We have amended the text to explicitly discuss the results for agriculture:

From Results

Moreover, the least-affected sector is, in each case, agriculture, which some might have expected to be especially sensitive to seasonal climate (although perhaps not to the winter climate targeted by the June ENSO and Winter Outlooks).

Referee 3: Specific Comment 8

Referee 3: Specific Comment 8

Figure 3a-b, I'm not sure "total market cap exposed" is the right terminology. Perhaps "total market cap of companies with some exposure"? Obviously there is no way an El Nino event could ever cause \$16 trillion of impacts. The regression results give the difference in risk before versus after the forecast release. But it is not clear how one could recover the baseline El Nino risk from the difference. In any event, it would seem more important to focus on the premium and risk reduction numbers rather than the market cap.

We agree with the suggested change in language. We have made that change throughout. And we have added the following text to our discussion:

From Results

As not all of a firm's value will be exposed to the seasonal climate, these numbers are loose upper bounds on the market capitalization exposed to each seasonal outlook.

Referee 3: Specific Comment 9

Referee 3: Specific Comment 9

Page 23, line 398, it would seem that the binomial tree model does not account for fat-tailed risk in securities prices. It might be helpful to point this out.

Thank you for the suggestion. We have added the following text:

From Methods

This pricing model can be seen as a generalization (and discretization) of the Black-Scholes pricing model to allow for early exercise and account for dividends. As such, it does not account for “fat-tailed” risks or jump processes.

Referee 4

Thank you for your helpful comments and guidance. We describe how we have addressed all of your concerns below.

Your comments are below, `boxed in with a gray background and different font`. My responses are in plain text. Any included text from the new manuscript is `boxed in with a white background and the` `same font as this text` with the section number noted at the top of the quoted text.

Referee 4: Overview

Referee 4: Overview

This paper is potentially a very important demonstration of the aggregate economic impact and value of seasonal forecasts, which previously has not been quantified except in comparatively limited case studies. In my opinion it will be a valuable and doubtless highly cited addition to this literature, although certain details need further addressing and/or explaining as outlined below.

I should emphasize before continuing that my perspective is that of a climate scientist with expertise in seasonal predictability and prediction, but not the financial data and analysis methods that are at the core of this study. Therefore it is essential that the paper also be reviewed (or have been reviewed) by someone familiar with the methods applied to assess the impact of anticipated information on option volatility, as exemplified in references 56-63.

That said, the simple illustrative example in Fig. 1 is nicely crafted and helpful to the non-specialist for understanding how seasonal outlooks potentially can influence option prices by reducing uncertainty.

Thank you for the kind evaluation. We are happy that you found the example helpful.

Referee 4: Specific Comment 1

Referee 4: Specific Comment 1

The presented results appear for the most part to be solid statistically, with some possible exceptions that are discussed below. However, as for many statistical analyses, various choices are made that potentially could channel the outcomes toward spurious statistical significance if the authors allowed themselves to be so influenced. On one hand, their arguments for the singular importance of the June ENSO outlook are plausible. On the other hand, it is not obvious why, for example, the authors focus on ‘‘an October–December target as that window is especially informative about boreal winter conditions’’ (lines 495–496), considering that (a) such a window mostly precedes boreal winter, (b) the sea surface temperature manifestations of ENSO tend to be strongest in November to January, and (c) ENSO’s impacts on North American climate tend to be most pronounced after December, in the mid-to-late winter and early spring months. Could the authors provide a more specific and robust argument for this choice, and comment on sensitivity to considering alternative targets such as November–January or December–February?

We understand that the referee asks us to conduct a sensitivity analysis with respect to the target we use to measure the skill of various months’ ENSO outlooks. In our prior submission, we used an October–December ENSO target on the advice of scientists who lead ENSO prediction efforts at NOAA. In response to the referee, we have added a new section to the Supplementary Information (labeled F) that assesses sensitivity to three different types of targets. The first two are targets requested by the referee: a November–January target and a December–February target. The third is a

constant five-month lead-time target. The new text in SI F reads as follows:

In the main text, we measure the ENSO Outlook's skill by the anomaly correlation coefficient for forecasts of an October-November-December (OND) target. Figures S-9 and S-10 assess sensitivity to a November-December-January (NDJ) target, a December-January-February (DJF) target, and a constant 5-month lead time target. The skill for each target is measured as the anomaly correlation coefficient from (17). The skill of each month's outlook varies across panels of a figure, but the estimated reductions in risk exposure and option market premia are constant across panels. We see that skill jumps from the May to the June outlook regardless of the forecast target, whereas there is very little change in skill between the June and July outlooks regardless of forecast target.

Table S-6 considers the implications for the value of a 1% improvement in ENSO prediction skill. The first column reports the anomaly correlation coefficients from (17). The June outlook's skill is similar across forecast targets, but the May outlook has greater skill at later forecast targets. If we based our value calculations on the later targets, then we would estimate greater value because a smaller jump in skill now explains the same jump in the market premium.

The remaining columns duplicate our value calculations with alternate forecast targets. The first row provides the values reported in the main text. The second and third rows show that using the jump in skill at forecasting an NDJ or DJF target would increase our estimated value of the June ENSO outlook by 30–40%. The final row shows that using skill at a constant 5-month lead time (OND for the May outlook and NDJ for the June outlook) yields results that are similar to those reported in the main text. The values reported in the main text appear conservative.

We copy the new Table S-6 as Table R2 below.

Table R2: Value of a 1% increase in ENSO skill.

	Outlook Skill		Value of 1% Skill Improvement	
	May	June	Risk Exposure (\$billion)	Option Market Premium (\$million)
OND Target	0.8193	0.8623	18.2 (-16.4,52.8)	1.79 (-0.31,3.90)
NDJ Target	0.8342	0.8676	23.9 (-21.5,69.3)	2.35 (-0.41,5.11)
DJF Target	0.8285	0.8591	25.9 (-23.3,75.1)	2.55 (-0.44,5.54)
5-Month Target	0.8193	0.8676	16.2 (-14.6,47.0)	1.60 (-0.28,3.47)

Skill measured as the anomaly correlation coefficient from (17).
95% confidence intervals in parentheses.

We have edited the main text to say,

From Results

Supplementary Information F shows that we estimate 30–40% larger effects of a 1% improvement in skill if we measure skill at forecasting either a November–January or December–February target instead of an October–December target.

And we have edited the Methods to say,

From Methods

Supplementary Information F assesses sensitivity to other forecast targets.

We thank the referee for suggesting these additional robustness checks.

Referee 4: Specific Comment 2

Referee 4: Specific Comment 2

Regarding the relative ordering of sectoral impacts on the June ENSO vs Winter Outlooks shown in Fig. 2c-d and discussed near line 125, would it be worth noting that ENSO impacts are global, whereas the Winter Outlook pertains to the United States, and that different sectors likely have different global vs national seasonal climate exposures?

Thank you for the suggestion. We have added the following sentence to that discussion:

From Results

These differences could reflect, among other differences, the global nature of ENSO impacts as opposed to the U.S. focus of the Winter Outlook.

Referee 4: Specific Comment 3

Referee 4: Specific Comment 3

The effects in Fig 3b-c associated with the CSU Hurricane Outlook are significant at $p < 0.05$, but opposite in sign to those associated with the June ENSO Outlook. Do the authors have any hypotheses for why this is, other than it being a statistical fluke?

This is a good question, thank you. There are two parts to our answer.

First, regarding the sign of the point estimate, Figure S-1 in SI E shows that the CSU hurricane outlook has zero effect on most sectors but does increase variance in several. Extrapolating these results to the firm level suggests that there happens to not be enough firms with randomly negative

effects on the variance of their options to offset the ones with randomly positive effects. We believe this is a statistical fluke.

Second, regarding the significance of the estimate, the confidence intervals in Figure 3 were underestimated in our original manuscript. We wrote in the Methods,

From Methods

Standard errors come from summing variances across firms and taking the square root. Tests suggest that the error introduced by ignoring correlation across each pair of firms' coefficient estimates is small.

This method estimated effects in separate regressions firm by firm and then, of necessity, treated firm estimates as independent (i.e., as if off-diagonals in a full covariance matrix were all zero) when aggregating over firms. We adopted this approximation for computational reasons, as the full calculation required estimating a very high dimensional regression. Spurred by the referee's comments, we have undertaken to recalculate confidence intervals in a fashion that accounts for potential correlation across firms. To do so, we now run a single regression with firm-specific coefficients rather than regressing firm by firm. Combining all in a single regression permits us to cluster standard errors by date, as in the paper's other specifications. We then use the full covariance matrix to calculate standard errors. Our revised Methods read,

From Methods

We estimate a version of (1) modified to permit the effects of a forecast release to vary by firm:

$$\ln (IV_{it}/IV_{i(t-1)}) = \beta_{fi0}D_{ft} + \beta_{f(-1)}D_{f(t-1)} + \beta_{f1}D_{f(t+1)} \\ + \Gamma_{iy}X_{it} + \epsilon_{it}.$$

We use the estimated $\hat{\beta}_{fi0}$ as described below. The point estimates resulting from the calculations described below are extremely similar if we instead obtain the $\hat{\beta}_{fi0}$ by estimating (1) firm by firm.

and

From Methods

We obtain the standard error for each forecast by the delta method, using the full covariance matrix of the $\hat{\beta}_{fi0}$.

It turns out that the combination of regressing all firms jointly with date clustering and accounting for off-diagonal elements of the covariance matrix did greatly expand the confidence intervals in Figures 3 and 4. Tests suggest that being able to cluster by date (rather than accounting for off-diagonals in the covariance matrix) is the key reason for the expanded confidence intervals. Nonetheless, the effects of the June ENSO Outlook in these figures are still significant at the 1% level. And pertinent to the referee's comment, the 95% confidence interval for the CSU outlook now reaches across zero (p=0.053). We appreciate the spur to figure out how to make the full calculations computationally feasible and hope that the revised calculations satisfy the referee.

Referee 4: Specific Comment 4

Referee 4: Specific Comment 4

Can the authors explain why the effects in Fig 4b are far more statistically significant than those in Figs 4a and c, particularly as this implies effects of the May and July outlooks that are hard to explain in terms of their hypotheses? Could these uncertainties perhaps have been systematically underestimated? (If so then this bears on Fig. 3b as well.)

These are each different calculations. Figure 4a is the average marketwide effect on implied volatility. Figures 4b and 4c run regressions with firm-specific effects and weight the results by firms' market capitalizations and option characteristics. These figures' results depend on how the weights covary with firms' regression coefficients. That said, the referee's hunch is correct: the confidence intervals formerly reported in Figures 4b and 4c were underestimated owing to an approximation we had made for computational reasons. See our reply to specific comment 3. In the revised manuscript, neither the May nor the July outlook is statistically different from zero, whereas the June outlook continues to be significantly different from zero.

Referee 4: Specific Comment 5

Referee 4: Specific Comment 5

Regarding Fig 4a, it is expected that 1 or 12 cases would be significant with $p \leq 0.1$ purely by chance. Therefore the (marginal) significance of this result is that it occurs for the June Outlook, in accord with the authors' hypothesis.

Thank you for raising this point. We make three changes in response. First, we clarify that the June outlook is significant at the 5% level by adding the parenthetical remark in the following sentence:

From Results

Consistent with the one-off jump in skill upon moving past the spring barrier, the June Outlook is the only release that shows significant effects at the 10% level (and it is significant at even the 5% level, $p = 0.013$).

Second, we have added new evidence in support of the referee's point:

From Results

In addition, Supplementary Information E shows that the June Outlook is one of only two outlooks (along with the May outlook) with a reduction in risk exposure that is significant at the 10% level (and it is significant at even the 1% level, $p = 0.0039$) and one of only two outlooks (along with the January outlook) with an option market premium that is significant at the 10% level (and it is again significant at even the 1% level, $p = 0.0025$).

Third, we then make the referee's point:

From Results

We would expect that 1 out of 12 tests would show significance at the 10% level purely by chance in each of these tests, but the consistently significant result for the June outlook is precisely the one that we predicted would occur if traders are sensitive to the well-known increase in skill around the spring barrier.

Referee 4: Specific Comment 6

Referee 4: Specific Comment 6

The ENSO outlook skill data that is described as being provided in reference 37 is presented there in a graphical form that is not readily translated into precise numerical values. Therefore the authors must have obtained the numerical data from the authors of that paper, or else have reproduced their calculations. Could this be clarified? (Possibly this is addressed in the Supplementary Information, which I did not have access to.)

Good question. One of the authors provided us with tabular data. We have clarified the origins of the data by editing the Methods to read:

From Methods

Using author-provided tabular data from ⁷⁴, ...

Referee 4: Specific Comment 7

Referee 4: Specific Comment 7

Reference 31 needs to be updated.

Good catch. We have fixed this reference to reflect that the cited paper is now published.

Referee 4: Specific Comment 8

Referee 4: Specific Comment 8

Some final comments: From the perspective of seasonal forecast providers, this study answers some questions but raises others, particularly in relation to the “scientific literacy” of the markets. Given the apparent economic advantages involved, it may not be surprising that the markets apparently are aware of the relatively large skill jump from the May to June ENSO outlook, for example, and can differentiate between more and less skillful and trustworthy outlooks which may have similar visibility. An interesting question would be how such knowledge is transmitted from the science to the financial communities. For example, is there a small number of climate experts in the financial world whose knowledge diffuses throughout the community to become widely known rules of thumb? Another aspect is that the official outlooks seldom hold surprises for knowledgeable members of the scientific and perhaps financial communities, who through monitoring and experience will be able to anticipate the broad character of the outlooks weeks to months before they are released. (This is broadly addressed in lines 208-214.) Finally, although the effects described in this paper align with widely accepted science, it is worth being alert to exceptions. One possible exception is that reference 20 found markets to have apparently priced in the Francis and Vavrus (2015) suggestion that anthropogenically-driven changes in the jet stream are causing an increased incidence of cold winters in the eastern US, given that this view has remained controversial.

Thank you for all of these perceptive, high-level thoughts. Anecdotally, we are aware that climate expertise had increased in recent years in financial firms both by experience of the second author (via her own career and those of scientists she has mentored) and by media reports on the importance of climate expertise in hedge funds. We have attempted to succinctly capture the main thrust of the referee's thoughts by adding the following sentence to the second paragraph of the Discussion:

From Discussion

Future work should investigate how other types of knowledge are transmitted from scientific communities to markets in order to understand the critical links.

REVIEWERS' COMMENTS

Reviewer #3 (Remarks to the Author):

None. My comments have been addressed.

Reviewer #4 (Remarks to the Author):

The authors have done a good job responding to and acting on my comments, and acceptance of the revised submission is recommended.

Just to add that I did notice one and possibly two inconsequential errors in the author's reply to my review:

1) p. 22 refers to Figs. S9-S10, but I believe what was meant was S10-S11

2) p. 22 regarding Table S6 states: "The second and third rows show that using the jump in skill at forecasting an NDJ or DJF target would increase our estimated value of the June ENSO outlook...". According to column 3 of Table S6 there is a slight jump in skill from the OND to NDJ target, however skill for the DJF target is slightly lower than either of these although the values in columns 4-5 are increased compared to the OND target as the authors state. Perhaps I misinterpreted the meaning of "jump in skill", although I couldn't see any alternative interpretation, such as the June minus May differences in skill for various targets, for which the quoted statement is valid.

These glitches in the response have no bearing on the revised submission; that they were noticed is hopefully indicative of my careful reading of the response.

Replies to Referees, Round 3

March 2024

Reviewer #3 had no comments.

Reviewer #4 wrote:

“

The authors have done a good job responding to and acting on my comments, and acceptance of the revised submission is recommended.

Just to add that I did notice one and possibly two inconsequential errors in the author's reply to my review:

1) p. 22 refers to Figs. S9-S10, but I believe what was meant was S10-S11

2) p. 22 regarding Table S6 states: “The second and third rows show that using the jump in skill at forecasting an NDJ or DJF target would increase our estimated value of the June ENSO outlook...”. According to column 3 of Table S6 there is a slight jump in skill from the OND to NDJ target, however skill for the DJF target is slightly lower than either of these although the values in columns 4-5 are increased compared to the OND target as the authors state. Perhaps I misinterpreted the meaning of “jump in skill”, although I couldn't see any alternative interpretation, such as the June minus May differences in skill for various targets, for which the quoted statement is valid.

These glitches in the response have no bearing on the revised submission; that they were noticed is hopefully indicative of my careful reading of the response.

“

We reply:

- 1) This typo in figure labels was where we had copied and pasted from the SI to the replies document. It did not apply to the SI itself, so there was no fix to make.
- 2) This comment was based on a misreading of the text in the same part of the SI. We have edited the language in that sentence to avoid a similar misreading (in particular, we have clarified that we are comparing the change in skill ****from May to June**** across the various outlook targets).